# The kleisin subunit controls the function of *C. elegans* meiotic cohesins by determining the mode of DNA binding and differential regulation by SCC-2 and WAPL-1

**Maikel Castellano-Pozo[1], Georgios Sioutas[1], Consuelo Barroso[1], Josh P Prince[1], Pablo Lopez-Jimenez[2], Joseph Davy[1], Angel-Luis Jaso-Tamame[1], Oliver Crawley[1], Nan Shao[1], Jesus Page[2], Enrique Martinez-Perez[1,3]\***

[1]MRC London Institute of Medical Sciences, London, United Kingdom; [2]Universidad Autónoma de Madrid, Madrid, Spain; [3]Imperial College Faculty of Medicine, London, United Kingdom

**\*For correspondence:**
enrique.martinez-perez@
imperial.ac.uk

**Competing interest:** The authors declare that no competing interests exist.

**Abstract** The cohesin complex plays essential roles in chromosome segregation, 3D genome organisation, and DNA damage repair through its ability to modify DNA topology. In higher eukaryotes, meiotic chromosome function, and therefore fertility, requires cohesin complexes containing meiosis-specific kleisin subunits: REC8 and RAD21L in mammals and REC-8 and COH-3/4 in *Caenorhabditis elegans*. How these complexes perform the multiple functions of cohesin during meiosis and whether this involves different modes of DNA binding or dynamic association with chromosomes is poorly understood. Combining time-resolved methods of protein removal with live imaging and exploiting the temporospatial organisation of the *C. elegans* germline, we show that REC-8 complexes provide sister chromatid cohesion (SCC) and DNA repair, while COH-3/4 complexes control higher-order chromosome structure. High-abundance COH-3/4 complexes associate dynamically with individual chromatids in a manner dependent on cohesin loading (SCC-2) and removal (WAPL-1) factors. In contrast, low-abundance REC-8 complexes associate stably with chromosomes, tethering sister chromatids from S-phase until the meiotic divisions. Our results reveal that kleisin identity determines the function of meiotic cohesin by controlling the mode and regulation of cohesin–DNA association, and are consistent with a model in which SCC and DNA looping are performed by variant cohesin complexes that coexist on chromosomes.

## Editor's evaluation

This landmark paper clarifies the distinct role of two meiosis cohesin complexes with different klesin subunits. With a temporally-resolved depletion method for a target protein combined with high-quality imaging in *C. elegans* meiosis, the authors provide convincing evidence to support their conclusions. This work will be of broad interest to colleagues in the fields of meiosis research as well as chromosome biology.

## Introduction

Cohesin belongs to a family of structural maintenance of chromosomes (SMCs) protein complexes that are essential components of mitotic and meiotic chromosomes due to their ability to modify

the topology of DNA (*Yatskevich et al., 2019*). In somatic cells, cohesin ensures correct chromosome segregation by providing sister chromatid cohesion (SCC) between S-phase and the onset of anaphase, ensures genome stability through its role in DNA damage repair, and regulates gene expression by controlling genome folding through DNA looping. In addition, during meiosis, cohesin orchestrates the formation of proteinaceous axial elements that promote pairing and recombination between homologous chromosomes and facilitates the two-step release of SCC during the consecutive meiotic divisions (*Klein et al., 1999*; *Grey and de Massy, 2021*). As inter-homologue recombination and the step-wise release of SCC are prerequisites for the formation of euploid gametes (*Petronczki et al., 2003*), the correct function of meiotic cohesin is essential for fertility.

The core of cohesin consists of a ring-shaped complex formed by two SMC proteins (Smc1 and Smc3) and a kleisin subunit that bridges the ATPase heads of Smc1 and Smc3. A family of HAWK (Heat repeat proteins Associated With Kleisin) proteins are recruited by the kleisin to control the loading, removal, and activity of cohesin on DNA (*Wells et al., 2017*). These include Scc3, Pds5, and the cohesin loader Scc2, which is required for the association of cohesin with mitotic and meiotic chromosomes (*Lightfoot et al., 2011*; *Ciosk et al., 2000*). Pds5 and Scc3 in turn interact with Wapl, a factor that promotes removal of cohesin throughout the cell cycle in a manner that allows reloading of removed complexes (*Kueng et al., 2006*). Cleavage of the kleisin subunit by separase at anaphase onset during the mitotic and meiotic divisions triggers irreversible cohesin removal and release of SCC to promote chromosome segregation (*Uhlmann et al., 1999*; *Buonomo et al., 2000*). Thus, the kleisin subunit plays a key role in controlling cohesin's action on DNA.

An essential aspect of meiotic chromosome morphogenesis is the loading of cohesin complexes in which the mitotic kleisin Scc1 is replaced by the meiosis-specific kleisin Rec8 (*Klein et al., 1999*; *Watanabe and Nurse, 1999*). Substitution of Rec8 by Scc1 during yeast meiosis results in severe defects during meiotic prophase and in premature loss of SCC during the first meiotic division (*Brar et al., 2009*; *Tóth et al., 2000*), evidencing that Rec8 cohesin is functionally different from Scc1 cohesin. Higher eukaryotes express additional meiosis-specific kleisins beyond REC8, including RAD21L in mammals and the highly identical and functionally redundant COH-3 and COH-4 in *Caenorhabditis elegans*, which could act as functional counterparts of mammalian RAD21L (*Severson et al., 2009*; *Gutiérrez-Caballero et al., 2011*; *Lee and Hirano, 2011*; *Ishiguro et al., 2011*; *Severson and Meyer, 2014*). How complexes differing in their kleisin subunit perform the multiple roles of meiotic cohesin remains poorly understood. Moreover, the dynamic association of cohesin with chromosomes is key for some of cohesin's functions in somatic cells (*Tedeschi et al., 2013*), but whether meiotic cohesin associates dynamically with chromosomes is not known. In this study, we exploit the experimental advantages of the *C. elegans* germline to investigate how REC-8 and COH-3/4 complexes contribute to different aspects of meiotic cohesin function and whether this involves dynamic association of these complexes with meiotic prophase chromosomes.

## Results and discussion

### Dynamically bound COH-3/4 complexes are the main organisers of axial elements

We first set up to determine the relative abundance of the three meiotic kleisins (REC-8, COH-3, and COH-4) on the axial elements of three-dimensionally intact pachytene nuclei. The endogenous *rec-8*, *coh-3*, and *coh-4* loci were individually tagged with GFP by CRISPR to generate three strains homozygous for GFP-tagged versions of each of the meiotic kleisins. GFP tagging of meiotic kleisins did not affect protein functionality as strains homozygous for *rec-8::GFP,* or for both *coh-3::GFP* and *coh-4::GFP* showed normal chiasma formation (*Figure 1A*). To quantify the signal intensity of meiotic kleisins in pachytene nuclei, germlines were dissected, stained with anti-GFP antibodies, and imaged under the same exposure conditions. Half-nucleus sum projections (sums pixel values of each XY coordinate in the Z series) were made to prevent overlap between axial elements from different chromosomes and to capture all signal associated with individual axial elements. This was followed by quantification of the fluorescent signal at the axis (*Figure 1B*). This demonstrated that COH-3 is the most abundant kleisin on pachytene axial elements, which contain similar amounts of COH-4 and REC-8 (*Figure 1B*). The ratio of COH-3/4 to REC-8 cohesin is 3.5, consistent with the observation that COH-3/4 cohesin plays a more prominent role than REC-8 cohesin in axis integrity of pachytene nuclei

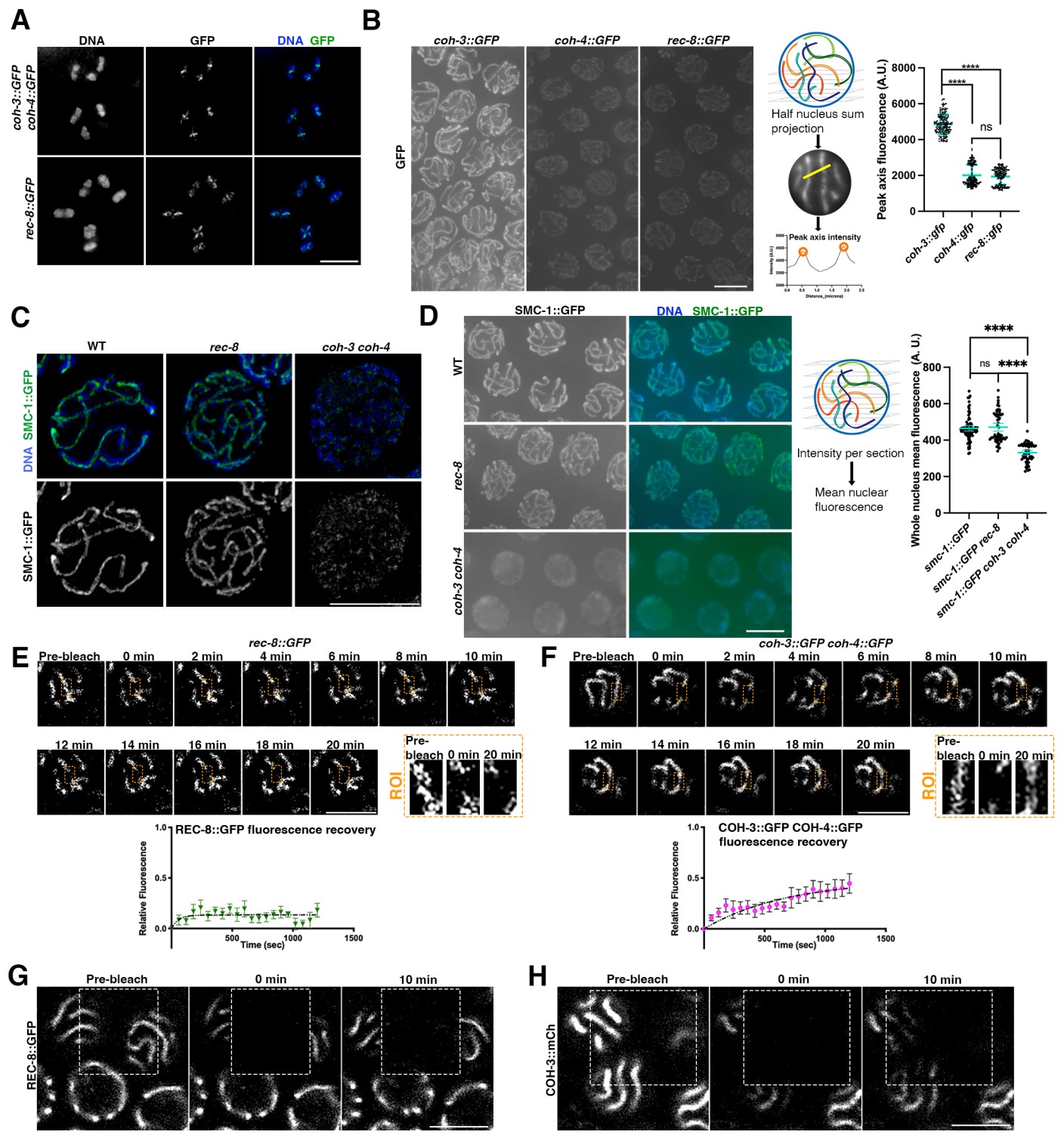

**Figure 1.** Highly abundant COH-3/4 complexes promote axis integrity and associate dynamically with pachytene chromosomes. (**A**) Projections of diakinesis oocytes stained with DAPI and anti-GFP antibodies to visualise COH-3::GFP COH-4::GFP (top panel) or REC-8::GFP (bottom panel). The presence of six bivalents confirms normal chiasma formation. (**B**) Non-deconvolved projections of pachytene nuclei of indicated genotype stained with anti-GFP antibodies. Images were acquired under the same exposure conditions and panels were adjusted with the same settings to allow direct comparison of anti-GFP staining intensity in the different genotypes. Diagram depicts creation of half-nucleus sum intensity projections to measure anti-GFP signal intensity at axial elements. Graph shows anti-GFP intensity quantification of sum intensity at axial elements, error bars indicate mean and SD, and p-values were calculated using a two-tailed Mann–Whitney *U* test. N (number of axes analysed [one or two per nucleus]: 192 [*coh-3::GFP*],

*Figure 1 continued on next page*

*Figure 1 continued*

200 [*coh-4::GFP*], 191 [*rec-8::GFP*]). Nuclei from three different germlines per genotype were included. (**C**) Projections of pachytene nuclei from worms expressing SMC-1::GFP of indicated genotypes stained with anti-GFP antibodies and DAPI and imaged by structural illumination microscopy (SIM). Note the presence of continuous-linear structures containing SMC-1::GFP in WT and *rec-8* mutants, but not in *coh-3 coh-4* double mutants. (**D**) Non-deconvolved projections of pachytene nuclei of indicated genotype stained with anti-GFP antibodies. Images were acquired under the same exposure conditions and panels were adjusted with the same settings to allow direct comparison of anti-GFP staining intensity in the different genotypes. Diagram depicts creation of whole-nucleus mean intensity projections to measure anti-GFP signal. Graph shows anti-GFP intensity quantification, error bars indicate mean and SD, and p-values were calculated using a two-tailed Mann–Whitney *U* test. N (number of nuclei analysed): 83 (*smc-1::GFP* [WT]), 59 (*smc-1::GFP rec-8*), and 60 (*smc-1::GFP coh-3 coh-4*). (**E, F**) Fluorescence recovery after photobleaching (FRAP) analysis of REC-8::GFP and COH-3::GFP COH-4::GFP in pachytene nuclei. Images show pre- and post-photobleaching images at indicated time points, with orange rectangles indicating the photobleached area on the axial element that was followed through the experiment. As nuclei move through the experiment, the focal plane was adjusted to follow the indicated region of the bleached axial element, while other regions can be out of focus. n = 9 nuclei (REC-8::GFP), n = 11 nuclei (COH-3::GFP COH-4::GFP). Error bars indicate SEM. (**G**) High-resolution FRAP images of worms expressing REC-8::GFP (genotype: *rec-8::GFP rec-8Δ*) at indicated times before and after photobleaching the area indicated by the dashed rectangle. Note that 10 min after photobleaching there is no recovery of REC-8::GFP signal on bleached axial elements. (**H**) High-resolution FRAP images of worms expressing COH-3::mCherry and REC-8::GFP (genotype: *coh-3::mCherry coh-3Δ rec-8::GFP rec-8Δ*) at indicated times before and after photobleaching the area indicated by the dashed rectangle. Note that 10 min after photobleaching there is recovery of COH-3::mCherry signal on bleached axial elements. Scale bar = 5 μm in all panels.

The online version of this article includes the following source data and figure supplement(s) for figure 1:

**Source data 1.** Source data for graphs in *Figure 1B, D, E and F*.

**Figure supplement 1.** SC staining in kleisin mutants and regions of interest used for FRAP analysis.

(*Castellano-Pozo et al., 2020*) and with previous estimations of REC-8 and COH-3/4 abundance on spread pachytene nuclei (*Woglar et al., 2020*). To explore the contribution of REC-8 and COH-3/4 cohesin to the organisation of axial elements, we first used structural illumination microscopy (SIM) to image SMC-1::GFP (a cohesin SMC subunit common to all types of cohesin in worms) in germlines of controls, *rec-8* single, and *coh-3 coh-4* double mutants. Continuous axial elements labelled by SMC-1::GFP were observed in wild-type controls and *rec-8* mutants, while SMC-1::GFP signals appeared as discontinuous weak signals in pachytene nuclei of *coh-3 coh-4* double mutants (*Figure 1C*), in agreement with previous observations using SMC-3 antibodies (*Severson and Meyer, 2014*). Next, we performed a quantitative analysis of the SMC-1::GFP signal in wild-type controls and kleisin mutants. Given the lack of continuous axial elements in *coh-3 coh-4* double mutants, we used whole-nucleus mean intensity instead of sum intensity at axial elements to quantify SMC-1::GFP signal in pachytene nuclei (*Figure 1D*). SMC-1::GFP signal intensity was similar in wild-type controls and *rec-8* mutants, while *coh-3 coh-4* double mutants displayed a clear reduction in signal intensity (*Figure 1D*), confirming that in pachytene nuclei most SMC-1 is associated with COH-3/4 cohesin complexes. The large differences in the abundance of REC-8 and COH-3/4 complexes in axial elements of pachytene nuclei can explain the discontinuous appearance of HORMAD proteins and synaptonemal complex components in *coh-3 coh-4* mutants (*Severson et al., 2009* and *Figure 1—figure supplement 1A*). Thus, by ensuring the integrity of axial elements, COH-3/4 complexes are key regulators of higher-order chromosome organisation during meiotic prophase.

We next addressed whether REC-8 or COH-3/4 complexes associate dynamically with meiotic prophase chromosomes by performing fluorescence recovery after photobleaching (FRAP) in pachytene nuclei of live animals. Following photobleaching of a small region of an axial element, we observed very little recovery of axis-associated REC-8::GFP signal, evidencing little reloading of REC-8::GFP over the imaging period (20 min) (*Figure 1E* and *Figure 1—figure supplement 1B*). In contrast, FRAP analysis of pachytene nuclei from *coh-3::GFP coh-4::GFP* homozygous worms demonstrated regaining of fluorescence signal on the photobleached region of the axis, which approached 50% of pre-bleach levels over 20 min (*Figure 1F*). To further confirm these observations, we also performed high-resolution FRAP experiments (see 'Methods') in which we bleached a larger area spanning more than one nucleus and acquired images at only three times points (to minimise bleaching caused by acquisition) in pachytene nuclei from transgenic strains expressing fully functional fluorescently tagged versions of REC-8 and COH-3 (*Castellano-Pozo et al., 2020*). Consistent with the full time-course experiment shown above, high-resolution FRAP demonstrated clear reloading of COH-3::mCherry, but not REC-8::GFP to axial elements of pachytene nuclei 10 min after photobleaching (*Figure 1G and H*). Thus, in contrast with REC-8 cohesin, COH-3/4 complexes associate dynamically with pachytene axial elements. In mammalian somatic cells, the creation of a stable pool of cohesin

requires passage through S-phase (*Gerlich et al., 2006*) and in worms only REC-8 complexes associate with chromosomes during meiotic S-phase, while COH-3/4 load post S-phase (*Severson and Meyer, 2014*). Therefore, the different loading time of REC-8 and COH-3/4 cohesin may be a determinant of the dynamics of these complexes. As RAD21L complexes also associate post S-phase during mouse meiosis (*Ishiguro et al., 2014*), our results suggest that, similar to COH-3/4, RAD21L may also associate dynamically with pachytene chromosomes. Elucidating the dynamics of REC8 and RAD21L during mammalian meiosis remains an important question.

## WAPL-1 and SCC-2 promote the dynamic association of COH-3/4 cohesin with pachytene chromosomes

The dynamic association of COH-3/4 cohesin with pachytene axial elements led us to test whether factors that promote loading and removal of cohesin during the mitotic cell cycle also control COH-3/4 dynamics. In mammalian somatic cells, WAPL is required to maintain a pool of cohesin that associates dynamically with chromosomes (*Tedeschi et al., 2013*), and we have previously shown that in the absence of WAPL-1 the levels of axis-associated COH-3/4 are increased (*Crawley et al., 2016*), suggesting that WAPL-1 may control the dynamic association of COH-3/4 complexes. To determine whether this was the case, we performed FRAP in *wapl-1* mutant worms. While the dynamics of REC-8::GFP were similar in the presence and absence of WAPL-1, showing almost no increase of post-photobleaching fluorescence intensity (*Figure 2A*), in the case of COH-3::GFP and COH-4::GFP the increase in post-photobleaching intensity was largely lost in the absence of WAPL-1 (*Figure 2A*). Therefore, WAPL-1 is required to ensure the dynamic association of COH-3/4 cohesin with axial elements in pachytene nuclei. WAPL also induces cohesin removal during late meiotic prophase in yeast and plants (*Challa et al., 2019*; *De et al., 2014*), but whether this also occurs at earlier meiotic stages is not currently understood. Removal of WAPL in mouse oocytes at the end of meiotic prophase results in increased binding of SCC1 cohesin, but not REC8 cohesin (*Silva et al., 2020*), suggesting that REC8 cohesin is also largely refractory to WAPL-mediated removal during mammalian meiosis. Thus, the establishment of a population of cohesin that is refractory to the removal activity of WAPL may be a conserved feature of the meiotic programme.

The fact that COH-3/4 cohesin complexes associate de novo with axial elements during pachytene led us to evaluate whether SCC-2, which is required for cohesin loading at the onset of meiosis and localises to pachytene axial elements (*Lightfoot et al., 2011*), controls this process. We first verified that SCC-2 is indeed required for the initial loading of COH-3/4 cohesin (*Figure 2—figure supplement 1A*) as previous studies only assessed this indirectly by monitoring SMC-1 loading (*Lightfoot et al., 2011*). As auxin-mediated protein degradation allows cohesin depletion from pachytene nuclei (*Castellano-Pozo et al., 2020*; *Zhang et al., 2015*), we tagged the endogenous *scc-2* locus with GFP and AID tags at its C-terminus using CRISPR and crossed this strain to worms expressing TIR1 (required for auxin-mediated degradation). Homozygous *scc-2::AID::GFP TIR1* worms displayed normal chiasma formation (*Figure 2—figure supplement 1B*), confirming SCC-2 functionality. We reasoned that if SCC-2 promotes reloading of COH-3/4 complexes that are removed by WAPL-1 (see below), then, depletion of SCC-2 following normal cohesin loading during early meiosis should induce a decrease of COH-3/4 cohesin in pachytene nuclei. As nuclei take about 36 hr to progress from meiotic S-phase to late pachytene (*Jaramillo-Lambert et al., 2007*), treating *scc-2::AID::GFP* homozygous worms with auxin for 8–14 hr should result in germlines containing mid and late pachytene nuclei that underwent meiotic S-phase in the presence of SCC-2 plus a population of early prophase nuclei that lacked SCC-2 from the onset of meiosis (*Figure 2B*). We confirmed that auxin treatment for 8 and 14 hr induced efficient depletion of SCC-2::AID::GFP from meiotic nuclei (*Figure 2B* and *Figure 2—figure supplement 1C*). Staining of auxin-treated germlines with anti-COH-3/4 antibodies showed that SCC-2 depletion for 8 hr induced a reduction of 35% of COH-3/4 signal intensity (quantified as whole-nucleus mean intensity) in mid pachytene nuclei that underwent meiotic S-phase before the start of auxin treatment (*Figure 2B* and *Figure 2—figure supplement 1D*). COH-3/4 signal intensity in mid pachytene nuclei was further reduced to 48% of untreated controls following 14 hr of auxin treatment (*Figure 2B* and *Figure 2—figure supplement 1D*). In contrast, SCC-2 depletion resulted in only a moderate decrease (9%) of REC-8 signal intensity in mid pachytene nuclei after 14 hr of auxin treatment (*Figure 2B*). As expected, SCC-2 depletion did eliminate REC-8 staining from axial elements of early prophase nuclei that underwent meiotic S-phase following auxin exposure, resulting

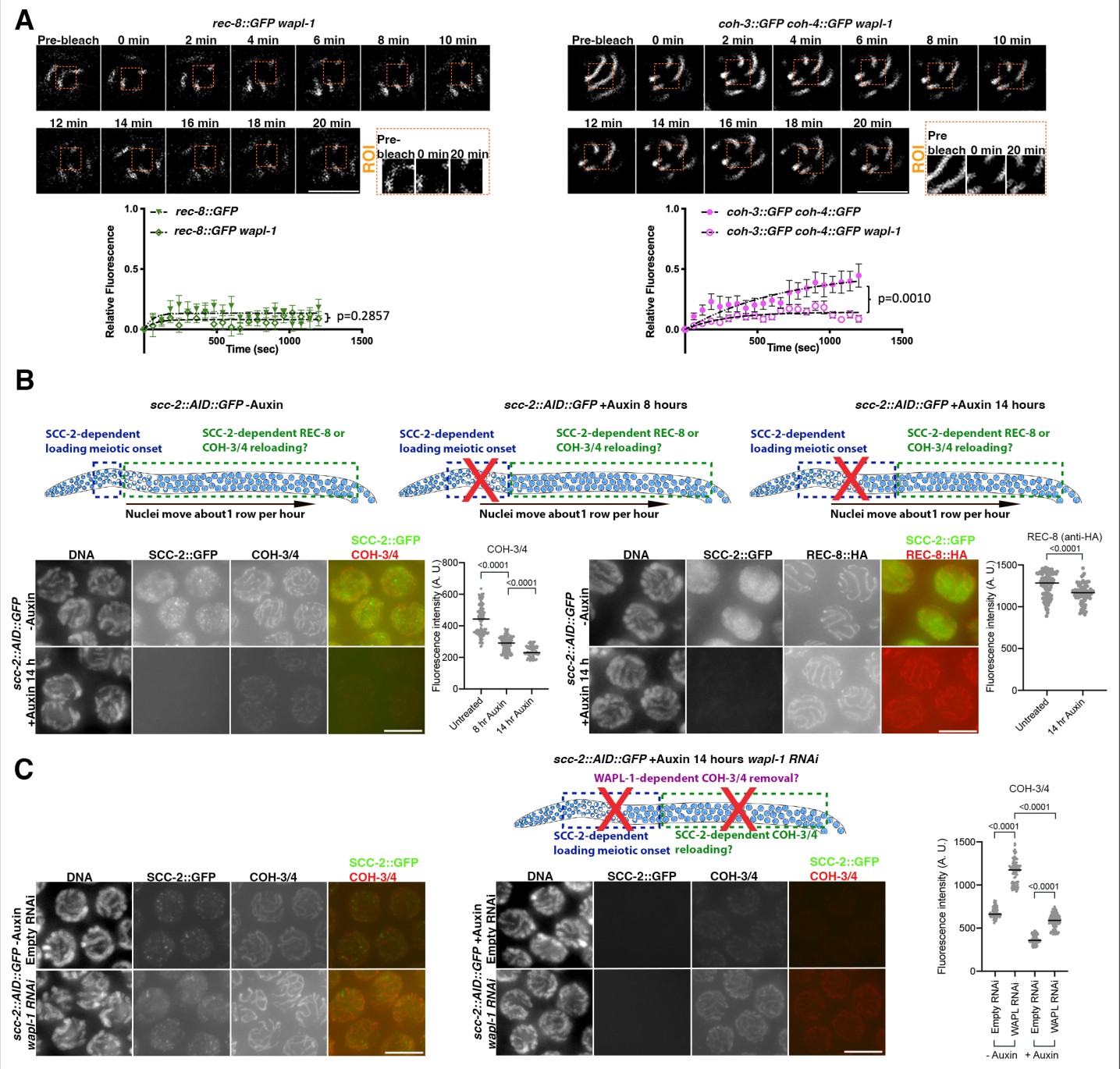

**Figure 2.** WAPL-1 and SCC-2 control the dynamic association of COH-3/4, but not REC-8, complexes with pachytene chromosomes. (**A**) Fluorescence recovery after photobleaching (FRAP) analysis of REC-8::GFP and COH-3::GFP COH-4::GFP in pachytene nuclei of WT and *wapl-1* mutants. Data for WT REC-8::GFP and WT COH-3::GFP COH-4::GFP is the same as shown in *Figure 1D*. Number of nuclei analysed: n = 9 (REC-8::GFP), n = 10 (REC-8::GFP *wapl-1*), n = 11 (COH-3::GFP COH-4::GFP), and n = 10 (COH-3::GFP COH-4::GFP *wapl-1*). Error bars indicate SEM, and p-values were calculated by Mann–Whitney tests between the Ymax predicted values of the one-phase association curves of individual experiments. (**B**) Effect of SCC-2 on REC-8 and COH-3/4 cohesin levels in pachytene nuclei. Diagrams on the top row of the panel indicate the population of nuclei affected by auxin-mediated SCC-2::AID::GFP depletion at different times of auxin exposure. Nuclei in regions marked by green rectangle underwent meiotic S-phase before auxin treatment, allowing the evaluation of a potential requirement of SCC-2 post S-phase. Projections of raw images from pachytene nuclei from *scc-2::AID::GFP* (left) and *scc-2::AID::GFP rec-8::HA* (right) from control worms (untreated) and from worms treated with 4 mM auxin for 14 hr. Nuclei from worms of the same genotype were acquired under the same exposure conditions and panels were adjusted with the same settings to allow direct comparison of -auxin and +auxin images. SCC-2::GFP was visualised using anti-GFP antibodies, COH-3/4 using anti-COH-3/4 antibodies, and REC-8 using anti-HA antibodies. As REC-8::HA and COH-3/4 were visualised with different antibodies, their relative staining intensity

*Figure 2 continued on next page*

*Figure 2 continued*

is not directly comparable, or comparable to those on *Figure 1B* in which REC-8, COH-3, and COH-4 were all tagged with GFP and visualised using anti-GFP antibodies. SCC-2::AID::GFP was efficiently depleted following auxin treatment, inducing strong decrease of COH-3/4 but not REC-8::HA signal. Graphs show quantification of whole-nucleus mean intensity in projections of pachytene nuclei before and after auxin treatment. Number of nuclei analysed: COH-3/4 (133 untreated, 193 +auxin 8 hr, 96 +auxin 14 hr) REC-8 (105 untreated, 89 +auxin), lines indicate median and p-values were calculated by two-tailed Mann–Whitney *U* test. (**C**) Effect of removing SCC-2::AID::GFP from germlines lacking WAPL-1. Diagram shows nuclei affected by SCC-2::AID::GFP depletion 14 hr after auxin treatment. Projections of raw images from pachytene nuclei from *scc-2::AID::GFP* of indicated treatments (+/-auxin, +/-*wapl-1* RNAi) acquired under the same exposure conditions and adjusted with the same settings to allow direct comparison of anti-COH-3/4 staining in all conditions. Quantification of whole-nucleus mean intensity shows that SCC-2 prevents WAPL-1-dependent and -independent COH-3/4 removal. Number of nuclei analysed: -auxin empty RNAi (79), -auxin *wapl-1* RNAi (80), +auxin empty RNAi (173), +auxin *wapl-1* RNAi (113), lines indicate median and p-values were calculated by two-tailed Mann–Whitney *U* test. Scale bar = 5 µm in all panels.

The online version of this article includes the following source data and figure supplement(s) for figure 2:

**Source data 1.** Source data for graphs in *Figure 2A–C*.

**Figure supplement 1.** Effect of SCC-2 depletion on pachytene nuclei.

in germlines containing REC-8-negative nuclei in early prophase followed by mid and late pachytene nuclei displaying strong REC-8 signal (*Figure 2—figure supplement 1E*). These findings show that SCC-2 is required for maintaining the levels of chromosome-bound COH-3/4, but not REC-8, cohesin in pachytene nuclei.

A possible explanation for our findings above is that the cohesin-releasing activity of WAPL-1 creates a pool of soluble COH-3/4 that is then loaded in an SCC-2-dependent manner onto axial elements of pachytene nuclei. We tested this hypothesis by inducing auxin-mediated removal of SCC-2 from pachytene nuclei of worms lacking WAPL-1 (*wapl-1 RNAi*), reasoning that in the absence of WAPL-1's removal activity the loading activity of SCC-2 may not be required to maintain COH-3/4 at pachytene axial elements (*Figure 2C*). As expected, *wapl-1* RNAi caused an increase in COH-3/4 levels, while auxin-mediated SCC-2 removal from pachytene nuclei of control worms (empty RNAi) caused reduced levels of COH-3/4 (*Figure 2C*). When SCC-2 was depleted from pachytene nuclei of *wapl-1* RNAi worms, COH-3/4 levels were not reduced compared to untreated (empty RNAi, no auxin) worms (*Figure 2C*), consistent with SCC-2 and WAPL-1 having antagonistic loading and unloading activities in pachytene nuclei. Interestingly, SCC-2 depletion from pachytene nuclei of *wapl-1* RNAi worms resulted in levels of COH-3/4 that were lower than those seen in nuclei from worms lacking just WAPL-1 (*wapl-1* RNAi, no auxin) (*Figure 2C*). This suggests that that in addition to promoting loading of COH-3/4 complexes removed by WAPL-1, SCC-2 may also act to prevent WAPL-1-independent removal of COH-3/4 complexes. Precedents for such mechanism have been reported in yeast, where Scc2 prevents removal of cohesin by a Wapl-independent mechanism during the G1 phase of the mitotic cell cycle (*Srinivasan et al., 2019*).

Our findings reveal that, beyond its requirement for cohesin loading at the onset of meiosis (*Lightfoot et al., 2011*), SCC-2 is a key regulator of cohesin dynamics in pachytene nuclei. Similar to worms, mouse NIPBL (SCC2) also localises to axial elements of pachytene nuclei (*Kuleszewicz et al., 2013*), suggesting that cohesin turnover during pachytene, presumably of non-cohesive RAD21L complexes, may be a conserved feature of meiosis. This possibility is further supported by observations in *Drosophila*, where the C(2)M kleisin, which does not provide SCC, is incorporated into axial elements during pachytene (*Gyuricza et al., 2016*). The localisation of SCC-2/NIPBL to pachytene axial elements may also indicate that active loop extrusion takes place at this stage as Scc2 is required for activating cohesin's ATPase activity and loop extrusion in vitro (*Davidson et al., 2019*; *Petela et al., 2018*), as well as for cohesin-mediated loop formation in mammalian cells during G2 (*Mitter et al., 2020*). Clarifying the roles of SCC-2 in controlling cohesin function during meiotic prophase is an important goal for future studies.

## SCC is provided by REC-8 complexes

We and others have proposed that during *C. elegans* meiosis SCC is exclusively provided by REC-8 cohesin, while COH-3/4 complexes associate with individual chromatids to regulate higher-order chromosome structure (*Woglar et al., 2020*; *Crawley et al., 2016*). However, this possibility is largely inferred from observations in mutants in which only REC-8 or COH-3/4 cohesin is present and therefore in which chromosome morphogenesis was partially impaired. Moreover, an involvement of COH-3/4

cohesin in SCC has also been proposed (*Severson and Meyer, 2014*) and in the absence of both WAPL-1 and proteins required for crossover formation COH-3/4 appear to mediate inter-sister attachments in diakinesis oocytes (*Crawley et al., 2016*). A direct assessment of the contribution of REC-8 and COH-3/4 cohesin to SCC under normal conditions requires the ability to specifically remove these complexes in a temporally resolved manner after normal chromosome morphogenesis. For example, in mouse oocytes arrested at metaphase I, TEV-mediated removal of REC8 revealed that SCC is exclusively provided by REC8 cohesin, despite the presence of SCC1 cohesin (*Tachibana-Konwalski et al., 2010*). We have previously shown that versions of REC-8::GFP and COH-3::mCherry carrying TEV recognition motifs in the central region of these kleisins are fully functional and allow kleisin cleavage following TEV protease micro injection into the germline (*Castellano-Pozo et al., 2020*). We took advantage of these strains to assess REC-8 and COH-3 contribution to SCC at different stages of meiosis. In the case of COH-3, experiments were performed in a *coh-4* mutant background due to functional redundancy between COH-3 and COH-4 (*Severson et al., 2009*). We first evaluated the impact of removing REC-8 or COH-3 from diakinesis oocytes, which contain six bivalents as a result of SCC and the presence of inter-homologue crossover events (*Figure 3A and B*). TEV-mediated removal of REC-8::GFP caused partial disassembly of diakinesis bivalents, which in most cases appeared as four lobed structures that remained weakly connected at the crossover site, with each lobe likely corresponding to one of the four chromatids present in a bivalent (*Figure 3A*). We called these structures 'lobed bivalents'. In addition to lobed bivalents, TEV-mediated REC-8 removal resulted in 35% of oocytes displaying individual chromatids that were fully detached (*Figure 3A*). A similar situation was observed when REC-8 was removed from diakinesis oocytes using auxin-mediated depletion (*Figure 3—figure supplement 1A*). Importantly, visualisation of COH-3 and condensin II confirmed that these SMC complexes remained bound to diakinesis chromosomes following REC-8 removal, including on fully detached chromatids (*Figure 3—figure supplement 1B and C*). Therefore, the observed disruption in SCC is specifically caused by the loss of REC-8 and not by an indirect effect of REC-8 removal on other SMC complexes.

The fact that COH-3 remains associated with fully detached chromatids following REC-8 removal in diakinesis oocytes (*Figure 3—figure supplement 1C*) suggests that COH-3/4 complexes associate with individual chromatids and therefore do not partake in SCC. Indeed, TEV-mediated removal of COH-3 or the mitotic kleisin SCC-1, which also associates with meiotic prophase chromosomes (*Severson and Meyer, 2014*), caused no obvious morphological changes in diakinesis bivalents (*Figure 3B* and *Figure 3—figure supplement 1D*), consistent with COH-3/4 or SCC-1 not providing SCC in diakinesis oocytes. We also attempted to simultaneously remove all cohesin variants by inducing auxin-mediated depletion of SMC-1, a subunit common to all cohesin complexes. SMC-1 depletion produced the appearance of lobed bivalents and detached chromatids in diakinesis oocytes (*Figure 3—figure supplement 1E*), as observed following REC-8 depletion. We noted that in most oocytes a small pool of SMC-1 signal remained associated with the crossover site (*Figure 3—figure supplement 1E*). We have previously observed that a population of REC-8 cohesin associated with crossover sites in late pachytene chromosomes is partially resistant to TEV-mediated removal (*Castellano-Pozo et al., 2020*). Thus, it is possible that a feature of chromosome structure present in the vicinity of crossover sites makes cohesin complexes associated with these regions more resistant to removal by the methods used here. To test whether a recombination-dependent feature could explain the weak inter-chromatid connections present in the lobed diakinesis bivalents induced by REC-8 removal, we removed REC-8 from diakinesis oocytes of *spo-11* mutants, which lack crossovers due to the absence of DNA double-strand breaks (DSBs) that initiate meiotic recombination (*Dernburg et al., 1998*). In this case, REC-8 removal induced a more penetrant loss of SCC (*Figure 3C*), suggesting that a recombination-dependent feature of chromosome organisation is involved in the weak chromatid connections observed when REC-8 is removed from diakinesis bivalents. We also tested a possible role of COH-3 in mediating SCC in diakinesis bivalents of *spo-11* mutants, but in this case, SCC remained intact following COH-3 removal (*Figure 3D*). These results suggest that SCC is exclusively provided by REC-8 complexes in diakinesis oocytes. This is consistent with the observation that the SPO-11-dependent tethering of sister chromatids observed in diakinesis oocytes of *rec-8* mutants represents inter-sister exchanges (*Crawley et al., 2016*; *Cahoon et al., 2019*; *Almanzar et al., 2021*), reinforcing that inter-sister attachments mediated by canonical SCC depend on REC-8 cohesin.

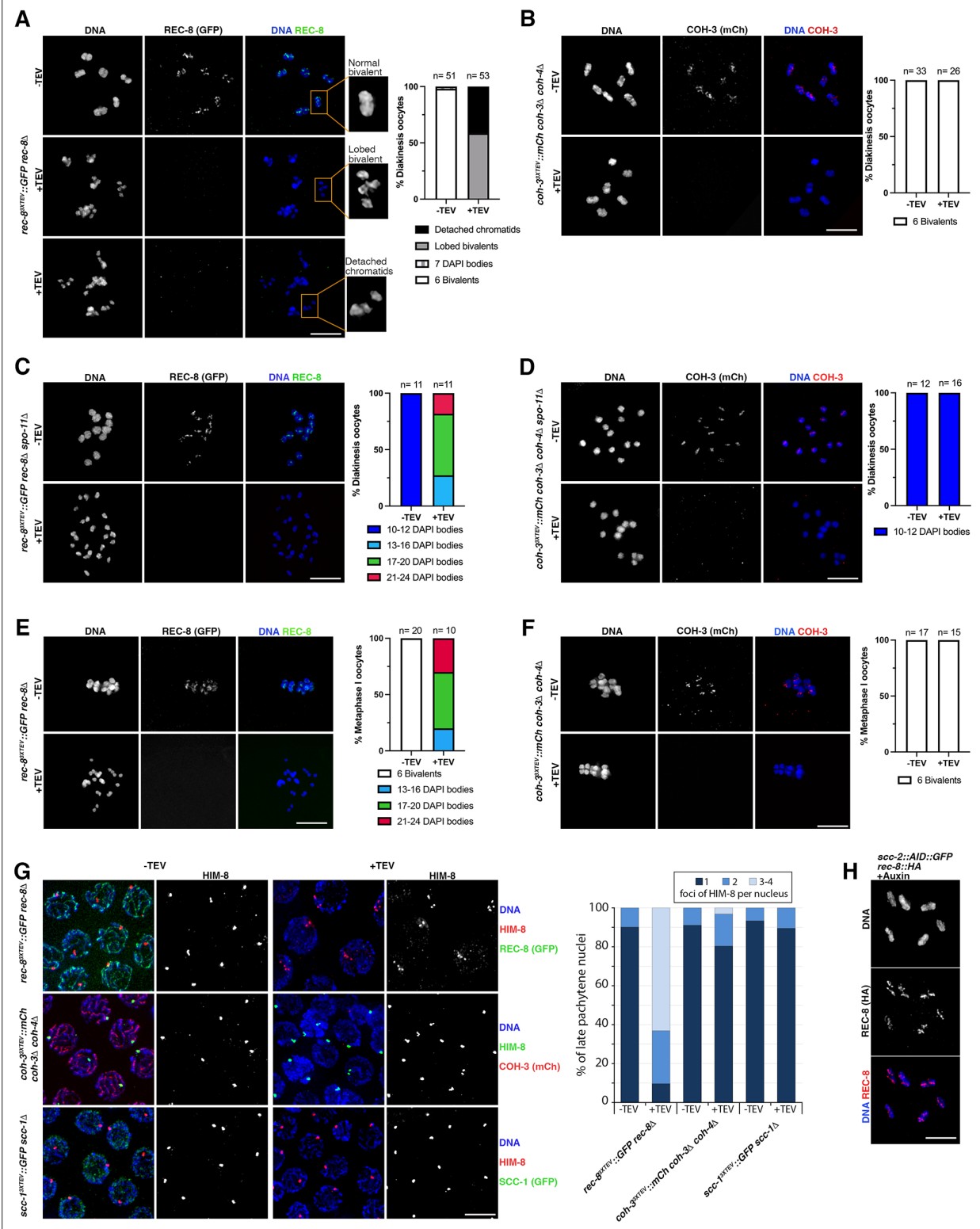

**Figure 3.** REC-8 complexes provide sister chromatid cohesion (SCC). (**A–D**) Diakinesis oocytes of indicated genotypes stained with DAPI and anti-GFP (REC-8) or anti-mCherry (COH-3) antibodies from untreated controls (-TEV) and 3.5 hr post TEV-mediated kleisin removal. Note that REC-8 removal transforms bivalents into lobed structures (middle panel in **A**) and also induces appearance of detached chromatids (bottom panel in **A**). Bivalent morphology remains unaffected by COH-3 removal (**B**). See magnified bivalents in (**A**) for examples of the different morphological categories used for the quantification of diakinesis oocytes. (**C, D**) REC-8, but not COH-3, removal induces separation of sister chromatids in *spo-11* mutant oocytes. (**E, F**)

*Figure 3 continued on next page*

*Figure 3 continued*

Metaphase I-arrested oocytes (*apc-2* RNAi) stained with DAPI and anti-GFP (REC-8) or anti-mCherry (COH-3) antibodies from untreated controls (-TEV) and 3.5 hr post-TEV-mediated kleisin removal. Note separation of sister chromatids following REC-8, but not COH-3, removal. (**G**) Late pachytene nuclei stained with DAPI, anti-HIM-8 antibodies, and anti-GFP (REC-8 or SCC-1) or anti-RFP (COH-3) antibodies from untreated controls (-TEV) and 3.5 hr following TEV-mediated kleisin removal. Note that removal of REC-8, but not COH-3 or SCC-1, leads to the appearance of nuclei with three or four HIM-8 foci, indicating separation of sister chromatids. Number of nuclei analysed = REC-8 (92 -TEV, 92 +TEV), COH-3 (135 -TEV, 128 +TEV), and SCC-1 (67 -TEV, 143 +TEV). (**H**) Diakinesis oocytes of *scc-2::AID::GFP rec-8::HA* worms exposed to auxin for 14 hr to induce SCC-2 depletion stained with DAPI and anti-HA (REC-8) antibodies. Note the presence of six bivalents displaying normal REC-8 staining and intact SCC. Scale bar = 5 µm in all panels.

The online version of this article includes the following source data and figure supplement(s) for figure 3:

**Source data 1.** Source data for graphs in *Figure 3A–G*.

**Figure supplement 1.** Effect of removing different cohesin subunits in diakinesis and metaphase I oocytes.

**Figure supplement 1—source data 1.** Source data for graphs in *Figure 3—figure supplement 1A, D, E, and F*.

Next, we focused our attention on SCC in oocytes at the metaphase I stage, when SCC is required to ensure correct orientation of bivalents on the spindle. Removal of REC-8 from metaphase I-arrested oocytes caused a general loss of SCC, with most oocytes displaying between 17 and 24 DAPI-stained bodies (full loss of SCC would result in 24 chromatids), while six bivalents remained present following COH-3 or SCC-1 removal (*Figure 3E and F* and *Figure 3—figure supplement 1F*). These results are consistent with REC-8 complexes providing SCC in metaphase I oocytes and suggest that the weak chromatid attachments that remain in diakinesis bivalents following REC-8 removal are resolved in metaphase I oocytes. This may be due to the assembly of the first meiotic spindle, which may be sufficient to pull apart weakly attached chromatids, or to the resolution of a crossover-associated chromosome structure between diakinesis and metaphase I. Regardless, the clear conclusion from our temporally resolved kleisin removal experiments is that, similar to mouse oocytes (*Tachibana-Konwalski et al., 2010*), SCC is exclusively provided by REC-8 cohesin in metaphase I oocytes of *C. elegans*.

Since SCC is established during meiotic S-phase and must persist until the meiotic divisions, and given that REC-8 provides SCC during metaphase I, we hypothesised that SCC is provided by REC-8 cohesin at all stages of meiotic prophase. We tested whether this was the case in pachytene nuclei by monitoring SCC at the chromosomal end of the X chromosomes bound by the HIM-8 protein (*Phillips et al., 2005*). Imaging of HIM-8 foci in late pachytene nuclei demonstrated that removal of REC-8, but not of COH-3 or SCC-1, caused loss of SCC (*Figure 3G*). These observations suggest that SCC is provided by REC-8 complexes in pachytene nuclei, consistent with a model in which complexes that provide SCC are stably bound to chromosomes between DNA replication and the onset of the meiotic divisions (*Burkhardt et al., 2016*). The stable association of cohesion-providing REC-8 complexes with meiotic chromosomes is also supported by our findings that SCC-2 is not required for maintaining REC-8 on pachytene chromosomes (*Figure 2B and E*). Moreover, we found that both REC-8 and SCC remained intact in diakinesis oocytes after 14 hr of auxin-mediated SCC-2 depletion (*Figure 3H*). In contrast, maintenance of normal levels of COH-3/4 complexes, presumably associated with individual chromatids, does require SCC-2 in pachytene nuclei (see *Figure 2B*). These differences between REC-8 and COH-3/4 in terms of their participation in SCC and their dependency on SCC-2 to sustain chromosomal association are reminiscent of observations in yeast mitotic cells, where Scc2 is essential for loading cohesin and maintaining cohesin's association with unreplicated DNA, but has no role in maintaining cohesion during G2/M phases (*Ciosk et al., 2000*; *Srinivasan et al., 2019*).

## REC-8 cohesin plays a more prominent role in DNA repair than COH-3/4 cohesin

Cohesin is thought to have diverse functions in the repair of DSBs, including an SCC-dependent role in homologous recombination (*Sjögren and Nasmyth, 2001*), and roles in regulating homology search of DSB ends and in the formation of repair foci, which are proposed to depend on the loop extrusion activity of cohesin (*Arnould et al., 2021*; *Piazza et al., 2021*). Our results so far show that stably bound REC-8 cohesin provides SCC while a larger population of dynamically associated COH-3/4 cohesin orchestrates chromosome organisation. We sought to clarify whether these differential activities correlate with the ability of REC-8 and COH-3/4 cohesin to promote DSB repair. Analysis of

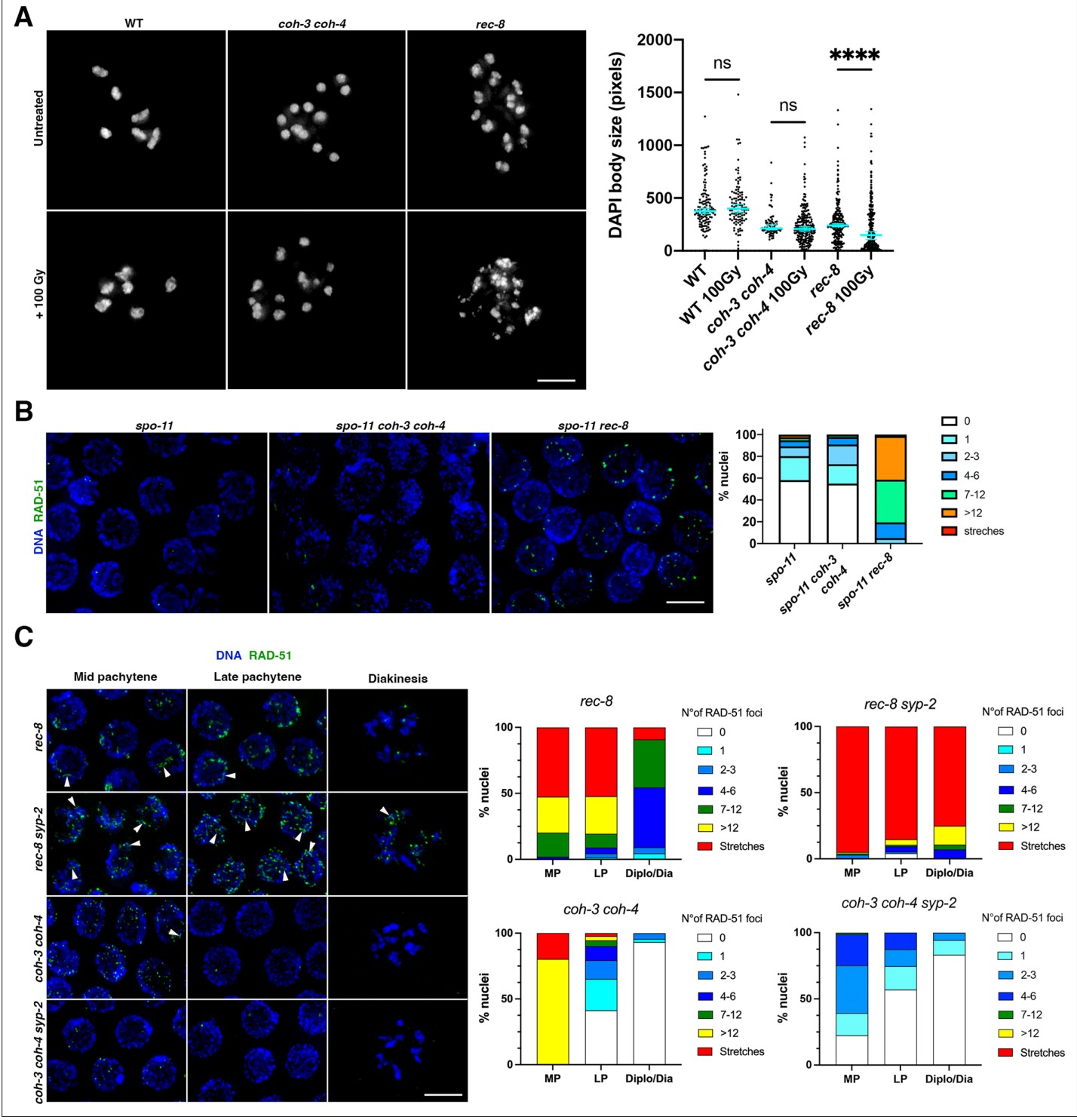

**Figure 4.** REC-8 complexes promote double-strand break (DSB) repair. (**A**) Diakinesis oocytes of indicated genotypes stained with DAPI in untreated controls and 26 hr after worms were exposed to 100 Gy of IR. Note the extensive appearance of small chromatin bodies in *rec-8* mutants, which indicate chromosome fragmentation. Graphs show the distribution of DAPI-stained bodies with a given area in full nucleus projections of diakinesis oocytes. Number of oocytes: WT n = 22 oocytes (0 Gy), n = 19 (100 Gy); *rec-8* n = 20 (0 Gy), n = 16 (100 Gy); *coh-3 coh-4* n = 7 (0 Gy), n = 22 (100 Gy). Error bars indicate median with 95% CI, p-values were calculated by two-tailed Mann–Whitney test. (**B**) Late pachytene nuclei of indicated genotypes stained with DAPI and anti-RAD-51 antibodies 24 hr after worms were exposed to 10 Gy of IR. Note high number of RAD-51 foci in *rec-8 spo-11*, but not in *spo-11 coh-3 coh-4* or *spo-11* mutants. Number of nuclei analysed = 298 (*spo-11*), 276 (*spo-11 coh-3 coh-4*), and 237 (*spo-11 rec-8*). (**C**) Nuclei of indicated genotypes stained with DAPI and anti-RAD-51 antibodies. Note much higher numbers of RAD-51 foci in *rec-8 syp-2* compared to *coh-3 coh-4 syp-2*.

*Figure 4 continued on next page*

*Figure 4 continued*

Arrowheads point to examples of elongated RAD-51 structures (stretches). Graphs show the percentage of nuclei with a given number of RAD-51 foci at the indicate stages (mid pachytene [MP], late pachytene [LP], diplotene/diakinesis [Diplo/Dia]) and genotypes. Number of nuclei analysed: *rec-8* (99 MP, 67 LP, 22 Diplo/Dia); *coh-3 coh-4* (102 MP, 92 LP, 45 Diplo/Dia); *rec-8 syp-2* (100 MP, 95 LP, 28 Diplo/Dia); *coh-3 coh-4 syp-2* (89 MP, 79 LP, 18 Diplo/Dia). Nuclei from three germlines per genotype were included in the quantification, nuclei from the –1 and –2 diakinesis oocytes were not included. Scale bar = 5 µm in all panels.

The online version of this article includes the following source data for figure 4:

**Source data 1.** Source data for graphs in *Figure 4A–C*.

chromosome morphology in diakinesis oocytes after exogenous DSBs are introduced by ionising radiation (IR) provides a clear read out of DSB repair capability, as impaired DSB repair during pachytene leads to chromosome fragmentation in diakinesis oocytes. Then, 26 hr after irradiation, chromosome fragments were largely lacking from diakinesis oocytes of wild-type worms exposed to 100 Gy of IR, evidencing efficient DSB repair (*Figure 4A*). In contrast, extensive chromosome fragmentation was evident in diakinesis oocytes of irradiated r*ec-8* mutants, with around 40% of the DAPI-stained bodies displaying a surface area consistent with chromosome fragments (*Figure 4A*). Chromosome fragmentation was also detected in diakinesis oocytes of irradiated *coh-3 coh-4* double mutants, although to a much lesser extent than in *rec-8* mutants. We also evaluated IR-induced DSB repair in pachytene nuclei by visualising recombination intermediates containing RAD-51 in backgrounds lacking SPO-11 and therefore endogenous DSBs. Then, 24 hr after exposure to 10 Gy, which induces a large accumulation of RAD-51 foci (*Lightfoot et al., 2011*), low numbers of RAD-51 foci were present in late pachytene nuclei of *spo-11* and *spo-11 coh-3 coh-4* mutants, consistent with efficient DSB repair (*Figure 4B*). In contrast, late pachytene nuclei of irradiated *rec-8* mutants contained high numbers of RAD-51 foci, evidencing impaired DSB repair (*Figure 4B*). Thus, SCC-providing REC-8 complexes appear to play a more prominent role in DSB repair than non-cohesive, dynamically bound, COH-3/4 complexes. Moreover, in pachytene nuclei of *rec-8* mutants, most sister chromatids are paired up due to the assembly of inter-sister synaptonemal complex (*Cahoon et al., 2019*), a phenomenon also observed in mouse *Rec8* mutants (*Xu et al., 2005*), which can promote inter-sister repair of SPO-11 DSBs (*Cahoon et al., 2019*; *Almanzar et al., 2021*). Supporting this possibility, double mutants lacking REC-8 and SC components display a striking accumulation of RAD-51 foci that persisted into diakinesis oocytes (*Cahoon et al., 2019*; *Figure 4C*). These late RAD-51 intermediates suggest the presence of unrepaired DSBs, which would explain the chromosome fragmentation seen in oocytes of mutants lacking the SC and REC-8 cohesin (*Crawley et al., 2016*; *Colaiácovo et al., 2003*). In addition to high numbers of regular size RAD-51 foci, *rec-8 syp-2* double mutant germlines also display elongated RAD-51 structures (stretches) (*Figure 4C*), suggesting the presence of abnormal recombination intermediates. In contrast to *rec-8 syp-2* double mutants, RAD-51 foci gradually decrease during late pachytene of *coh-3 coh-4 syp-2* triple mutant germlines and are largely lacking in diplotene and diakinesis oocytes (*Figure 4C*), consistent with the lack of chromosome fragments in oocytes of these mutants (*Crawley et al., 2016*). Thus, removing the SC from mutants lacking COH-3/4 cohesin does not compromise DSB repair, suggesting that inter-sister DSB repair mediated by REC-8 cohesin is sufficient to repair SPO-11 DSBs that accumulate in the absence of SC. Therefore, although COH-3/4 complexes are much more abundant than REC-8 complexes and are essential for the integrity of axial elements (*Figure 1B and C* and *Figure 1—figure supplement 1A*), SCC-providing REC-8 complexes play a much more prominent role in the repair of endogenous (SPO-11-dependent) and exogenous (IR) DSBs during late meiotic prophase. Our results suggest that similar to yeast Scc1 cohesin during mitotic G2 (*Sjögren and Nasmyth, 2001*), the role of REC-8 in DSB repair during meiotic prophase is mechanistically coupled to its ability to provide SCC.

## Conclusions

We reveal a clear distribution of functions between stably bound and low abundance REC-8 complexes, which provide SCC and DSB repair, and high-abundance COH-3/4 complexes that ensure the integrity of axial elements and associate dynamically with pachytene chromosomes in a process controlled by WAPL-1 and SCC-2. Our studies suggest functional conservation between worm and mouse REC8 cohesin, with these complexes providing SCC and being largely refractory to WAPL-mediated removal in both organisms (*Crawley et al., 2016*; *Silva et al., 2020*; *Tachibana-Konwalski et al., 2010*).

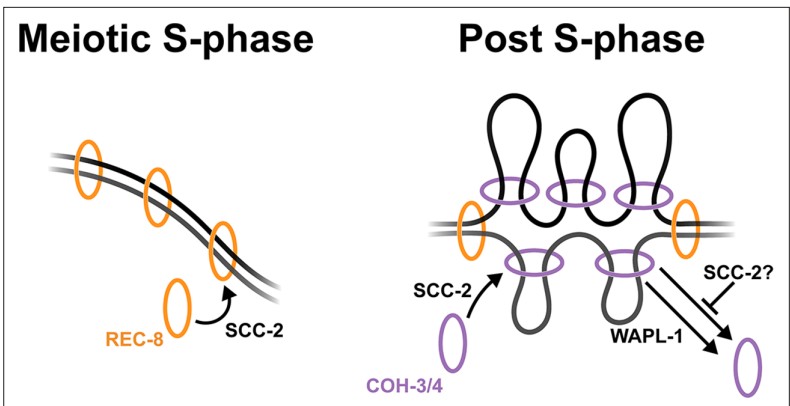

**Figure 5.** Model of functional specialisation of REC-8 and COH-3/4 cohesin. SCC-2-dependent loading of REC-8 cohesin during S-phase establishes sister chromatid cohesion. Non-cohesive COH-3/4 cohesin associates dynamically with individual chromatids post S-phase to control higher-order chromosome organisation in a process mediated by the loading and removal activities of SCC-2 and WAPL-1. SCC-2 may also act to prevent WAPL-1-independent removal of COH-3/4 cohesin.

Moreover, as we provide evidence suggesting that functional conservation may also extend to COH-3/4-RAD21L complexes, we hypothesise that, similar to *C. elegans* COH-3/4, mammalian RAD21L complexes may associate dynamically with meiotic chromosomes in a manner regulated by WAPL and NIPBL. Our studies support a model in which REC-8 complexes loaded during meiotic S-phase tether sister chromatids until the meiotic divisions, while COH-3/4 loaded post S-phase on individual chromatids act to control higher-order chromosome structure, likely by performing loop extrusion (*Figure 5*). Mitotic yeast and human cohesin complexes containing the Scc1 kleisin mediate both SCC and loop extrusion, but these processes are thought to involve different modes of DNA–cohesin interactions and are therefore proposed to be mutually exclusive (*Srinivasan et al., 2019*; *Davidson et al., 2019*). How different populations of Scc1 cohesin are regulated to perform either SCC or loop extrusion is not understood, but our findings suggest that during meiosis in higher eukaryotes these activities can be determined by kleisin identity.

## Methods

### *C. elegans* genetics and growing conditions

All strains were maintained on *Escherichia coli* (OP50) seeded NG agar plates at 20°C under standard conditions. N2 Bristol strain was used as the wild-type strain. All studies were performed using young adults at 18–24 hr post L4-larvae stage unless indicated. The following mutant alleles were used: *wapl-1(tm1814)*, *rec-8(ok978)*, *coh-3(gk112)*, *coh-4(tm1857)*, *scc-1(ok1017)*, *syp-2(ok307)*, and *spo-11(ok79)*. *Table 1* contains a full list of the strains used in this study. *C. elegans* strains generated in this study are available from the corresponding author.

### Generation of transgenic *C. elegans* strains

For the generation of transgenic strains carrying single copy insertions of the desired transgene, we used a strain carrying the *ttTi5605* MosSCI transposon insertion on chromosome II, except for transgene *fqSi18*, which was inserted at the *ttTi4348* locus on chromosome I. Transgene insertion was performed using the protocol described in *Frøkjær-Jensen et al., 2008*. The *scc-1^{3XTEV}::GFP* transgene was generated by adding a 75 bp fragment encoding for three repeats of the TEV recognition motif (ENLYFQGASENLYFQGELENLYFQG) after SCC-1's E293 codon in a vector expressing SCC-1::GFP under the *scc-1* promoter and 3′ UTR. *Table 2* contains a list of transgenes used in this study.

CRISPR-mediated genome editing was performed as described in *Paix et al., 2017* using preassembled Cas9-sgRNA complexes, single-stranded DNA oligos as repair templates, and *dpy-10* as a co-injection marker. To generate the *scc-2::AID::GFP* allele, we introduced 105 bp encoding the 35 amino acids of the AID tag (*Zhang et al., 2015*) between the last codon of *scc-2* and the start codon of GFP. The *rec-8::HA* allele was generated by introducing an 81 bp fragment encoding for three

**Table 1.** *C. elegans* strains used in this study.

| Strain | Genotype | Origin |
|---|---|---|
| ATG473 | *rec-8(syb803 [rec-8::GFP]) IV* | This study |
| ATG472 | *coh-3(syb751 [coh-3::GFP]) V* | This study |
| ATG556 | *coh-4(syb1273 [coh-4::GFP]) V* | This study |
| ATG707 | *coh-4(syb1273 [coh-4::GFP]) coh-3(syb751 [coh-3::GFP]) V* | This study |
| ATG228 | *smc-1 fq20 [smc-1::GFP] I* | Crawley |
| ATG252 | *smc-1 fq20 [smc-1::GFP] I; rec-8(ok978) IV / nT1 [qIs51] (IV;V)* | This study |
| ATG253 | *smc-1 fq20 [smc-1::GFP] I; coh-4(tm1857) coh-3(gk112) V / nT1 [qIs51] (IV;V)* | This study |
| ATGSi23 | *fqSi23 II; rec-8 (ok978) IV* | *Crawley et al., 2016* |
| ATGSi191 | *fqSi18 I; fqSi23 II; rec-8 (ok978) IV; coh-3 (gk112) V* | This study |
| ATGSi521 | *fqSi25 scc-1(ok1017) II* | This study |
| ATG571 | *wapl-1(tm1814) rec-8(syb803 [rec-8::GFP]) IV /nT1 [unc-? (n754) let-? qIs50] (IV;V)* | This study |
| ATG570 | *wapl-1(tm1814) IV; coh-3(syb751 [coh-3::GFP]) V /nT1 [unc-? (n754) let-? qIs50] (IV;V)* | This study |
| ATG572 | *wapl-1(tm1814) IV; coh-4(syb1273 [coh-4::GFP]) coh-3(syb751 [coh-3::GFP]) V //nT1 [unc-? (n754) let-? qIs50] (IV;V)* | This study |
| ATG282 | *scc-2(fq23 [scc-2::AID::GFP]) II; ieSi38 IV* | This study |
| ATG693 | *scc-2(fq23 [scc-2::AID::GFP]) II; ieSi38 rec-8(fq169[rec-8::3XHA]) IV* | This study |
| ATGSi355 | *fqSi16 II; rec-8(ok978) IV* | *Castellano-Pozo et al., 2020* |
| ATGSi441 | *fqSi15 II; coh-4(1857) coh-3(gk112) V* | *Castellano-Pozo et al., 2020* |
| ATGSi392 | *fqSi16 II; rec-8(ok978) spo-11(ok79) IV / nT1 [qIs51] (IV;V)* | This study |
| ATGSi470 | *fqSi15 II; spo-11(ok79) IV; coh-4(1857) coh-3(gk112) V / nT1 [unc-? (n754) let-? qIs50] (IV;V)* | This study |
| ATG323 | *fqSi17 II; ieSi38 rec-8(ok978) IV* | *Castellano-Pozo et al., 2020* |
| ATG541 | *rec-8(fq32[rec-8::TEV]) IV; coh-3(syb751[coh-3::GFP])* | This study |
| ATG415 | *smc-1 (fq64[smc-1::AID::GFP]) I; ieSi38 IV* | *Castellano-Pozo et al., 2020* |
| TY5120 | *coh-4(tm1857) coh-3(gk112) V/nT1 [qIs51] (IV;V)* | CGC |
| VC666 | *rec-8(ok978) IV/nT1 [qIs51] (IV;V)* | CGC |
| AV106 | *spo-11(ok79) IV/nT1 [unc-? (n754) let-?] (IV;V)* | CGC |
| ATG137 | *rec-8(ok978) IV; spo-11(ok79) IV/ nT1 [qIs51] (IV;V)* | *Crawley et al., 2016* |
| ATG213 | *spo-11(ok79) IV; coh-4(tm1857) coh-3(gk112) V/ nT1 [unc-? (n754) let-?] (IV;V)* | *Crawley et al., 2016* |
| ATG83 | *rec-8(ok978); syp-2(ok307)/nT1 [unc-? (n754) let-? qIs50] (IV;V)* | This study |
| ATG186 | *syp-2(ok307) coh-4(tm1857) coh-3(gk112)/nT1 [unc-? (n754) let-? qIs50] (IV;V)* | *Crawley et al., 2016* |

repeats of the HA tag (YPYDVPDYAYPYDVPDYAYPYDVPDYA) before *rec-8*'s STOP codon, while the *rec-8³ˣᵀᴱⱽ* allele was generated by introducing a 75 bp fragment encoding for three repeats of the TEV recognition motif (ENLYFQGASENLYFQGELENLYFQG) after *rec-8*'s Q289 codon. The *rec-8::GFP*, *coh-3::GFP*, and *coh-4::GFP* alleles were generated by SunyBiotech and all contained a C-terminal GFP sequence containing three artificial introns.

## Auxin-mediated protein degradation

All strains used for auxin-mediated protein degradation were homozygous for the *ieSi38* transgene (*Table 2*) expressing the TIR1 protein under the *sun-1* promoter and all proteins targeted for auxin-mediated degradation were expressed fused to the 35 amino acid AID tag (*Zhang et al., 2015*). Auxin treatment was performed by placing young adult worms, at 18–24 hr post L4-larvae stage, in seeded NG agar plates containing 4 mM auxin for the indicated periods of time.

**Table 2.** Transgenes used in this study.

| Transgene | Genotype |
|---|---|
| *fqSi23* | [*Prec-8 rec-8::GFP 3'UTR rec-8; cb-unc-119(+)*] |
| *fqSi18* | [*Pcoh-3 coh-3::mCherry 3'UTR coh-3; cb-unc-119(+)*] |
| *ieSi38* | [*Psun-1 TIR1::mRuby 3'UTR sun-1; cb-unc-119(+)*] |
| *fqSi16* | [*Prec-8 rec-8$^{3XTEV}$::GFP 3'UTR rec-8; cb-unc-119(+)*] |
| *fqSi15* | [*Pcoh-3 coh-3::$^{3XTEV}$::mCherry 3'UTR coh-3; cb-unc-119(+)*] |
| *fqSi17* | [*Prec-8 rec-8::AID::GFP 3'UTR rec-8; cb-unc-119(+)*] |
| *fqSi25* | [*Pscc-1 scc-1$^{3XTEV}$ 3'UTR scc-1; cb-unc119(+)*] |

## TEV protease microinjection

Germline injections were performed as described in *Castellano-Pozo et al., 2020* using a Narishige IM-31 pneumatic microinjector attached to an inverted Olympus IX71 microscope. Needles were made using a micropipette puller P-97 (Intracell) and borosilicate glass filaments with a 1.0 mm O.D. and 0.58 mm I.D. (BF100-58-10, Sutter Instruments). AcTEV Protease (Thermo Fisher, Cat# 12575) was used in a mix containing 10 U/µl TEV protease in 50 mM Tris-HCl, pH 7.5, 1 mM EDTA, 5 mM DTT, 50% (v/v) glycerol, 0.1% (w/v) Triton X-100.

## Immunostaining and image acquisition

Germlines from young adult hermaphrodites were dissected, fixed, and processed for immunostaining as described in *Castellano-Pozo et al., 2020*. Briefly, worms were dissected in EGG buffer (118 mM NaCl, 48 mM KCl$_2$, 2 mM CaCl$_2$, 2 mM MgCl$_2$, 5 mM HEPES) containing 0.1% Tween and fixed in the same buffer containing 1% paraformaldehyde for 5 min. Slides were immersed in liquid nitrogen before removing the coverslip and then placed in methanol at –20°C for 5 min, followed by three washes of 10 min each in PBST (1× PBS, 0.1% Tween) and blocking in PBST 0.5% BSA for 1 hr. Following incubation overnight with primary antibodies diluted in PBST, slides were washed three times for 10 min each in PBST. Slides were then incubated in the dark at room temperature for 2 hr with secondary antibodies diluted in PBST, washed with PBST three times for 10 min each and counterstained with DAPI. Finally, slides were washed for 1 hr in PBST and mounted using Vectashield (Vector). Images were acquired as 3D stacks using a ×100 lens in a Delta Vision Deconvolution system equipped with an Olympus 1X70 microscope, and images were deconvolved using SoftWoRx 3.0 (Applied Precision) and mounted using Photoshop.

## Super-resolution structured illumination microscopy

For images shown on *Figure 1C*, immunostaining was performed as described above, but slides were mounted using ProLong Diamond mounting media and covered with Zeiss high-performance 0.17 ± 0.005 coverslips. Images were acquired using a Zeiss Elyra S1 microscope, processed with Fiji, and mounted in Photoshop.

## Primary antibodies used

The following primary antibodies and dilutions were used: goat anti-GFP-488-conjugated (1:200) (Roche), rat anti-mCherry (1:1000) (5F8, ChromoTek), rabbit anti-COH-3/4 (1:400) (*Crawley et al., 2016*), mouse anti-REC-8 (1:100) (Novus Biologicals), guinea pig anti-HCP-6 (1:400) (*Chan et al., 2004*), rabbit anti-HIM-8 (1:500) (Novus Biologicals), and rabbit anti-RAD-51 (*Das et al., 2022*).

## Scoring number of DAPI-stained bodies in diakinesis and metaphase I oocytes

Worms of indicated genotypes and treatments were dissected and processed for immunostaining as described in the main methods, including the final DAPI staining step. Images were acquired as 3D stacks using a ×100 lens in a Delta Vision Deconvolution system equipped with an Olympus 1X70 microscope. Images were deconvolved using SoftWoRx 3.0 (Applied Precision) and mounted

in Photoshop. The number and appearance of DAPI-stained bodies in diakinesis and metaphase I oocytes were scored in 2D projections of three-dimensional intact germlines/embryos. Projections of diakinesis and metaphase I oocytes result in some overlap of DAPI-stained bodies, especially in situations where loss of SCC results in high numbers of DAPI-stained bodies; therefore in some cases, the number of DAPI-stained bodies may represent a slight underestimation of the overall number of individual chromatin bodies present.

The area in pixels of DAPI-stained bodies in diakinesis oocytes (*Figure 4A*) was calculated by generating maximum intensity projections of diakinesis nuclei and using CellProfiler to obtain the size in pixels of individual DAPI-stained bodies identified in 2D projections.

## Gamma irradiation

Worms were irradiated in a IBL 637 cell irradiator containing a caesium-137 source. NGM plates, containing young adult worms, were directly irradiated for the appropriate amount of time, resulting in irradiation of 100 Gy or 10 Gy, as required.

## Metaphase I arrest by RNA interference (RNAi) of *apc-2*

Metaphase I arrest was achieved by downregulation of the anaphase promoting complex (APC) via RNAi feeding using the *apc-2* clone from the Ahringer library (HT115 bacteria transformed with a vector for IPTG-inducible expression of dsRNA). Bacteria containing the *apc-2* vector, as well as empty vector (HT115) control, were both grown overnight at 37°C in LB with 50 µg/ml ampicillin. Cultures were collected and seeded onto NGM agar plates containing 1 mM IPTG and 25 µg/ml ampicillin. Plates were incubated overnight at 37°C to induce the expression of dsRNA. Then, 18–24 hr post-L4 worms were placed and experiments were performed after 48 hr.

## RNAi interference of *wapl-1*

*wapl-1* RNAi was performed by feeding worms with *E. coli* (HT115) containing a vector for IPTG-inducible expression of *wapl-1* RNAi from the Ahringer Library using an empty vector as negative control. Bacteria carrying the *wapl-1* and empty vectors were grown overnight at 37°C in 20 ml of LB with 100 µg/ml of ampicillin, cultures were then spun and resuspended in 1 ml of LB. Approximately 100 µl of bacteria were seeded onto NGM agar plates containing IPTG (1 mM) and ampicillin (100 µg/ml). After incubation at 37°C overnight, L3 animals were transferred onto the plates and allowed to lay F1 embryos that grew in these plates until the young adult stage. At 24 hr post-L4, F1 worms were moved to fresh agar plates containing IPTG (1 mM), ampicillin (100 µg/ml), and 4 mM of auxin to achieve auxin-mediated protein degradation of *scc-2*. Before dissection, animals were incubated for 14 hr in auxin plates and then analysed cytologically.

## Quantitative analysis of fluorescence intensity

To compare the occupancy of GFP-tagged meiotic kleisin subunits (*Figure 1B*), the peak axis fluorescence was measured. Dissected gonads of strains carrying GFP-tagged versions (generated by CRISPR) of REC-8, COH-3, and COH-4 were stained with FITC-conjugated aGFP antibodies. Acquisition was carried out on the Deltavision microscope, with a set exposure time. Sum-projections of the raw images were made on ImageJ to only include the half top or bottom of each nucleus (six-slice projections). Every nucleus was examined separately. To measure the peak axis intensity, a line was drawn over at least two clear axes of the nuclei, and the line profile was generated using the built-in 'Plot profile' function of ImageJ. Peak values were called using an online-available macro (found here:). The macro used was the 'Intensity' macro (Maxima and Minima of line profile Tool). Fluorescence measurements (in arbitrary units) were collected and the raw values were used to compare the different tagged protein occupancy by calculating the relative ratio of the proteins on the meiotic axis.

We used whole-nucleus mean fluorescence intensity to compare the immunostaining intensity of (1) SMC-1::GFP (using anti-GFP antibodies) in different genetic backgrounds (*Figure 1D*), (2) COH-3/4 (using anti-COH-3/4) antibodies in experiments testing the effect of SCC-2 and WAPL-1 (*Figure 2B and C*), and (3) REC-8::HA (using anti-HA antibodies) in experiments in which SCC-2::AID::GFP was depleted by auxin treatment (*Figure 2B*). Images were acquired on a Delta Vision microscope as 3D stacks using the same exposure settings in auxin-treated and untreated controls. For comparing fluorescence levels, non-deconvolved images were analysed in ImageJ. Nuclei of interest were manually

circled using the 'oval' tool, one nucleus at a time, and the fluorescence of each slice was measured. The mean fluorescence of that nucleus was then calculated, after normalising for the number of z-stack slices and the area of the circle drawn. Normalised, mean fluorescence values were directly compared between control and mutant strains, as arbitrary units.

### Fluorescence recovery after photobleaching (FRAP)

The FRAP method used here was modified from *Nadarajan et al., 2017* and *Pattabiraman et al., 2017*. Young adult worms were used for all FRAP experiments and immobilised on 5% agarose pads for imaging, covered in a solution of 2 mM levamisole in M9 medium. Imaging was performed in a Leica TCS SP5 system using the FRAP wizard (LASAF software), which allows for photobleaching of the regions of interest (ROIs). The FRAP wizard was only used for the pre-bleach, photobleaching and the 0 min post-bleach images. Images were taken in 2D every minute using the ×60 oil immersion lens, as 3D stack images were found to increase photobleaching in this setup. Nuclei of mid-pachytene were selected so that at least two axes were visible at the top of the gonad, where resolution was better. ROIs for photobleaching were designed to span at least two axes, with the ROI size being kept similar throughout experiments. One or two nuclei were bleached for every gonad, but only one nucleus per gonad was analysed for curve fitting. For acquisition, the following settings were used: 100 Hz scanning speed, two times line averaging, 1 AU pinhole. A 488 Argon laser was used at 30% power and 15% sub-power, and a HyD filter was used, set at 502–552 nm range. Bleaching was performed with the same laser at 60% sub-power, for 100 ms. Images were exported as .lif files and were analysed using ImageJ (version 2.0.0-rc-59/1.51k). The ImageJ built-in StackRreg algorithm was used to align the frames of the time series.

Aligned time-series images were analysed as described in *Nadarajan et al., 2017* and *Pattabiraman et al., 2017*. In short, three ROIs were designed: bleached area of the axis (ROI 1), the whole nucleus (ROI 2), and a background control, defined as a region in the gonad, but far enough from the bleached area (ROI 3). Intensity measurements were carried out for all ROIs using ImageJ, and data was analysed using Microsoft Excel. The analysis involved background subtraction (ROI 3) from ROI 1 and ROI 2. The subtracted values were then normalised against each other, so that fluorescence loss in ROI 2 was accounted for in ROI 1 (adapted from *Phair et al., 2004*). The ROI 1 post-bleach values were also normalised as relative intensities of the pre-bleach ROI 1 values. The resulting 'double-normalised' values were finally normalised again, this time by setting the initial post-bleach value to 1, by dividing all values by the initial difference (*Stenoien et al., 2001*).

Curve fitting was performed in GraphPad Prism software. Statistical analysis was performed using the one-phase association predicted $Y_{max}$ values (where the fitted curve is expected to plateau) and comparing them between different genotypes using the Mann–Whitney statistical test.

High-resolution FRAP images (*Figure 1G and H*) were acquired on a Leica TCS SP5 system with the following settings: 100 Hz scanning speed, three times line averaging, 1 AU pinhole. REC-8::GFP was imaged with 488 laser at 30% overall power and 10% sub-power for acquiring, HyD detector set to 502–552 nm range at 300 gain. COH-3::mCherry was imaged with a 561 laser and 10% imaging power, HyD detector set to 590–680 nm range, at 350 gain. Some manual adjustment of Z and X-Y focus was needed due to nuclear movement over long time points, which was done using very low-intensity imaging (1400 Hz) to minimise acquisition bleaching. Images were exported as TIFFs.

## Acknowledgements

We thank D Dormann and C Whilding from the MRC LMS microscopy facility for help with analysis of FRAP, Nicola Silva for providing anti-RAD-51 antibodies, and Barbara Meyer for providing anti-HCP-6 antibodies. This work was supported by an MRC core-funded grant to EMP, postdoctoral Fellowships from Fundación Alfonso Martin Escudero and EMBO to MCP, a grant from Universidad Autónoma de Madrid to JP and PLJ, and an EMBO scientific exchange grant to PLJ.

## Additional information

### Funding

| Funder | Grant reference number | Author |
| --- | --- | --- |
| Medical Research Council | MC-A652-5PY60 | Enrique Martinez-Perez |
| European Molecular Biology Organization | Postdoctoral Fellowship | Maikel Castellano-Pozo |
| Fundacion Alfonso Martin Escudero | Postdoctoral Fellowship | Maikel Castellano-Pozo |
| European Molecular Biology Organization | Scientific exchange grant | Pablo Lopez-Jimenez |
| Universidad Autonoma de Madrid | | Jesus Page Pablo Lopez-Jimenez |

The funders had no role in study design, data collection and interpretation, or the decision to submit the work for publication.

### Author contributions

Maikel Castellano-Pozo, Funding acquisition, Investigation, Methodology, Writing – original draft, Writing – review and editing; Georgios Sioutas, Oliver Crawley, Investigation, Methodology, Writing – review and editing; Consuelo Barroso, Pablo Lopez-Jimenez, Angel-Luis Jaso-Tamame, Nan Shao, Investigation, Writing – review and editing; Josh P Prince, Joseph Davy, Data curation, Formal analysis, Investigation, Writing – review and editing; Jesus Page, Supervision, Funding acquisition, Writing – review and editing; Enrique Martinez-Perez, Conceptualization, Supervision, Funding acquisition, Writing – original draft, Writing – review and editing

### Author ORCIDs

Maikel Castellano-Pozo  http://orcid.org/0000-0003-4134-9025
Josh P Prince  http://orcid.org/0000-0003-0877-7538
Pablo Lopez-Jimenez  http://orcid.org/0000-0002-6673-5996
Jesus Page  http://orcid.org/0000-0001-8381-324X
Enrique Martinez-Perez  http://orcid.org/0000-0001-5813-0383

### Decision letter and Author response

Decision letter https://doi.org/10.7554/eLife.84138.sa1
Author response https://doi.org/10.7554/eLife.84138.sa2

## Additional files

### Supplementary files

• MDAR checklist

### Data availability

All data generated or analysed during this study are included in the manuscript and supporting files; source data for all graphs on the manuscript are provided in source data files associated with each figure containing graphs.

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
