## [Editor Report]

This landmark paper clarifies the distinct role of two meiosis cohesin complexes with different klesin subunits. With a temporally-resolved depletion method for a target protein combined with high-quality imaging in *C. elegans* meiosis, the authors provide convincing evidence to support their conclusions. This work will be of broad interest to colleagues in the fields of meiosis research as well as chromosome biology.

---

## [Decision Letter]

**Decision letter after peer review:**

Thank you for submitting your article "The kleisin subunit controls the function of meiotic cohesins by determining the mode of DNA binding and differential regulation by SCC-2 and WAPL-1" for consideration by *eLife*. Your article has been reviewed by 3 peer reviewers, including Akira Shinohara as the Reviewing Editor and Reviewer #1, and the evaluation has been overseen by Detlef Weigel as the Senior Editor.

Essential revisions:

The reviewers acknowledge the importance of the results in the paper, which shows distinct dynamics of two cohesins in *C. elegans* meiosis. However, some data presented in the Figures are not yet sufficiently convincing to support the authors' conclusions. In addition, the description of the experiments including the controls is insufficient to judge the validity of the results. Therefore, several points need to be addressed. Since there are several major concerns, we may ask for a second round of review on the revised version.

1. Images in Figures 1C, 1D, 1E, 1F, 2B, and 2C should be replaced with more proper ones.

Figure 1C: The claim that axial elements stain continuously is not substantiated in the figure. Additional staining of an axis protein such as HTP-3 will be checked.

Figure 1D: Examples for differences in REC-8 and COH-3/4 recovery are not convincing. If no convincing example can be found, the analysis can't be performed.

Figures 1E and 1F: Signals outside of dotted squares are also decreased after photo-bleaching.

Figure 2B and 2C: SC-2(GFP) images had a high background in Figure 2B, but not in the bottom right image.

Also, images in the control experiments (no photobleaching in Figure 1 and staining in wild type without the tags) and other representative images should be provided in supplement or data storage sites. For image quantification such as intensity measurement, it is critical to show how to normalize the intensity between strains, in some cases with a high background signal, e.g. in Figures 1B 2B, and 2C, COH-4 and REC-8 show two distinct populations for the fluorescence. In Figures 2B, and C, a high background made it hard to quantify the intensity.

In addition, the differences between COH-3/4 staining between Figure 1 and Figure 2 are enormous (ca 7x), suggesting that SCC-2::AID is strongly sub-functional. This issue needs to be addressed at the minimum, or better eliminated by using a construct that behaves more like SCC-2.

2. For the AID experiments (Figure 2), it is critical to understand how fast SCC-2 is depleted and whether depletion is complete and therefore, for how long the depletion lasts in the various cell types. If possible, please show how much depletion of the protein was achieved by western blotting. Is the loading of COH-3/4 dependent on SCC-2? A gonad, like that in Figure 2C for REC-8, could be shown and analyzed for COH-3/4 as well.

3. The results showing the role of COH-3/4 in meiotic DSB repair are very weak at present. Additional experiments for DAPI and RAD-51 focus counting in spo-11 coh-3/4 with irradiation could help, introducing a second mutation (e.g. rad-54), which stalls any repair to eliminate the possibility of "rapid" repair.

4. One concern about the manuscript is how much different the contents of the paper are from the authors' previous publication (Crawley et al. *eLife* 2014; Castellano-Pozo et al. Nature Commun, 2020). It is essential to clarify what is new in this paper and what is confirmation of the previous works (or maybe correction of the work).

*Reviewer #1 (Recommendations for the authors):*

The paper elegantly reveals the role of REC-8 (but not of COH-3/4) in sister chromatid cohesion (SCC) in diakinesis and metaphase I. Importantly, REC-8 coheisin binds to meiotic chromosomes stably while COH-3/4-cohesin is more dynamic, probably accompanied by dissociation and association, which is controlled by cohesin regulators, SCC-2 and WAPL-1. The experiments in this paper were mostly conducted in a good shaper and most of the conclusions were supported by the data. The paper is clearly written and the conclusion is straightforward. On the other hand, there are some concerns about the data in the paper, which should be addressed prior to publication.

1. The results shown by FRAP experiments (Figures 1 and 2A) were not convincing. In *C. elegans* meiosis, chromosomes show motion driven by microtubule-dynein, which clearly affects the recovery of FRAP signals. It is best to perform FRAP in the absence of the motion in the Dynein mutants such as the dlc-1 mutant. Alternatively, the authors must argue that the motion did not affect their conclusion in a logical manner. Moreover, Figures 1E and 1F show peculiar behavior of signals. In the outside of the dotted square, GFP/mCherry signals diminished and GFP signals were recovered in Figure 1E (0 versus 10 min).

2. Figure 2A analyzed the effect of the wapl-1 (Wapl) mutation on Coh-3 dynamics and showed that the Wapl-1 promotes the re-association of Coh-3/4 on chromosomes. However, the authors did not discuss this in a reasonable way. Given that Wapl-1 promote the dissociation of the COH-3/4 cohesin from meiotic chromosomes (previously shown by the authors' group), this result suggests either that Wapl-1 has a new function to promote the association of the cohesin to the chromosomes or that WAPL-1 could remove COH-3/4 cohesin for its re-association. This should be discussed or addressed by analyzing wapl-1 scc-2 double mutants.

3. Since the authors had nice antibodies against COH-3/4 for immuno-staining some experiments should be done in double-staining for REC-8::HA and COH-3/4 in as Figure 2B right and Figure 2C-in Figure 2C. Moreover, in addition to Figure 2B let, it is essential to show the localization of COH-3/4 under SCC-2 depletion in "whole" gonads like in Figure 2C.

4. If the WAPL-1 controls the localization of COH-3/4, but not of REC-8, GFP-intensity measurement should show more COH-3/4-GFP signals in the wapl-1 mutant than the control, but the same amount of REC-8-GFP by measuring the intensity as shown in Figure 1B. This should be shown.

5. It is nice for the authors to show a more detailed structure of lobed bivalent in Figure 3A, by analyzing the structure with a super-resolution fluorescent microscope. For readers, it is nice to show a schematic drawing of the structure.

6. Given the contribution of cohesin in the formation of meiosis-specific chromosome axis structure, the authors showed and/or mentioned the role of REC-8 and COH-3/4 in axis structure by looking at the localization of axis proteins such as HORMD proteins.

7. Reproducibility: In some experiments such as Figure 2B, multiple nuclei in a single worm were measured. It is critical to show the results for independent worms.

*Reviewer #3 (Recommendations for the authors):*

Technical:

Figure 1

Figure 1D is not convincing at all.

REC-8::GFP – The object that is bleached is discontinuous with the other chromosomes. The fact that it disappeared is therefore no evidence that it was bleached without subsequent recovery. It is not clear what it was in the first place.

COH-3/4::GFP – Here the bleaching of a part of a chromosome was clearly successful, but the pictures do not convince me of recovery after 10 or even 20min. Instead, there is a substantial background that obscures a possible slight recovery.

The analysis seems not useful, if it is so difficult, to find even one "representative" case. If the authors manage to repeat the experiment more convincingly, it might be helpful to measure values for the background and its changes over time separately (dispersed signal between the actual chromosomes axes).

Figure 1 E. Only a part of the nucleus should be bleached to study redistribution of label – however at 0 min after bleaching REC-8::GFP to the left outside of the squared box is dramatically bleached. This suggests, that the full nucleus was bleached instead of just a part.

This in turn dramatically disturbs the equilibration kinetics.

The fact that the signal outside the bleaching field is actually back at 10 min is in contrast to the conclusions the authors derive.

If the signal came back, this would speak for clear dynamics of the REC-8 protein.

In addition, the chromosomes to the right of the bleaching area have a different shape – raising the possibility of movement, or an altered focal plane.

(see also typo on Y-axis label)

line 147: " the focal plane was adjusted to follow the indicated region of the bleached axial element, while other regions can be out of focus."

I think it may be difficult to focus on the bleached element, which is invisible because it is bleached. If the nuclei move very strongly it may be impossible to do meaningful FRAP without a control that allows to correct for the movement.

Figure 1 F. Similar to E the full nuclei seem to have been bleached. Again chromosome parts outside the bleaching area recover.

In summary, the documentation of the bleaching experiments is not convincing. I concede, that it is difficult.

Figure 2

My main problem with Figure 2 is that COH-3/4 untreated is now weak, punctate, or patchy and nothing like COH-3/4 in Figure 1B.

The authors write that COH-3/4 is 3.5 times stronger than REC-8 (l95) – yet COH-3/4 axes appear weaker than REC-8 axes and discontinuous.

The quantification shows an intensity average of 600 for COH-3/4 and double (1200) for REC-8. This varies by 2x3.5=7 fold from the statement on (l95).

The issue needs to be addressed properly. Should scc-2::AID cause the (7 fold)! reduction of COH-3/4 levels on chromatin, then scc-2::AID is a strongly sub-functional scc-2 version. It would be ideal, to change to a construct with better performance, but at the very minimum, those constitutive scc-2::AID defects should be directly addressed and described.

Also, SCC-2 staining does suffer from high background. Therefore it is close to impossible to judge the amount of SCC-2 after depletion. If cells are treated with auxin for 14 hours, secondary effects may arise.

It would be helpful to perform Western blotting, demonstrating how late after the addition of auxin SCC-2 disappears, and to which levels it is reduced. At the same time, it can be used as a control to test whether overall cellular levels of COH-3/4 and REC-8 are

unaffected at least by the scc-2::AID construct in the absence of auxin. Unfortunately, the moving window shown in Figure 2C right panel, cannot be captured by such an analysis.

Figure 3

312-314 „Since SCC is established during meiotic S-phase and must persist until the meiotic divisions, and given that REC-8 provides SCC during metaphase I, we hypothesised that SCC is provided by REC-8 cohesin throughout the meiotic prophase. We tested this possibility by monitoring SCC at the chromosomal end of the X chromosomes bound by the HIM-8 protein"

The HIM-3 experiment tests, whether REC-8 is required to maintain SCC on the X chromosome at some point in prophase if TEV is induced at some point in prophase.

It does not show the requirement „throughout" prophase.

Figure 4

398-400 „Chromosome fragments were largely lacking from diakinesis oocytes of wild-type worms exposed to 100 Gy of IR, evidencing efficient DSB repair (Figure 4A)."

Could the author indicate, at which time after irradiation the observation was made in diakinesis? How long did the cells have time to repair? What was the stage of the cells during irradiation?

The experiment shown in Figure 4 does not make a very strong additional point for a direct role of REC-8 in DSB repair.

Fragmentation after irradiation is an expected consequence after the loss of SCC, thus if REC-8 depletion results in loss of SCC this observation per se could be explained by the above-described defect of REC-8.

4A is complemented by the experiment in Figure 4B, which shows that the kinetics of disappearance of RAD-51 foci are at least slower in rec-8 mutants. Again though, this could be a direct consequence of the lack of SCC delaying homology search, rather than proving a more direct role of REC-8 in DSB repair.

Conclusions

435-438 The abundance shown in Figure 1 is not reflected in Figure 2. This creates some uncertainty about this relationship.

Is the scc-2::AID sub-functional?

---

## [Author Response]

Essential revisions:The reviewers acknowledge the importance of the results in the paper, which shows distinct dynamics of two cohesins in *C. elegans* meiosis. However, some data presented in the Figures are not yet sufficiently convincing to support the authors' conclusions. In addition, the description of the experiments including the controls is insufficient to judge the validity of the results. Therefore, several points need to be addressed. Since there are several major concerns, we may ask for a second round of review on the revised version.1. Images in Figures 1C, 1D, 1E, 1F, 2B, and 2C should be replaced with more proper ones.Figure 1C: The claim that axial elements stain continuously is not substantiated in the figure. Additional staining of an axis protein such as HTP-3 will be checked.

We have added two new figures in response to comments on Figure 1C. First, new Figure 1D shows raw images (acquired with the same settings) of SMC-1::GFP staining in WT, *rec-8*, and *coh-3 coh-4* double mutants confirming that continuous axial are observed in WT and *rec-8* mutants but not in *coh-3 coh-4* double mutants. Quantification of signal intensity in the three genotypes also confirms that most SMC-1::GFP signal in WT nuclei comes from complexes containing the COH-3/4 kleisins. Second, as requested, new Figure S1A shows staining of axial elements (HORMAD protein HIM-3) and SC (SYP-1) in WT, *rec-8*, and *coh-3 coh-4* double mutants. These experiments confirm that continuous axial elements and SC staining are present in WT and *rec-8* mutants, but not in *coh-3 coh-4* double mutants.

We have also added changes to Figures 1D-F and 2A-C, please see below for details on specific changes and additions to those figures.

Figure 1D: Examples for differences in REC-8 and COH-3/4 recovery are not convincing. If no convincing example can be found, the analysis can't be performed.

We now provide larger images clearly indicating the bleached axis at all time points, as well as a zoom in image of this region at pre-bleach, bleach, and 20 min post bleach. The COH-3::GFP COH-4::GFP example is a different nucleus to the one shown on the previous version. All panels are adjusted with the same settings to facilitate comparison of images at different time points. We think that these images clearly show reloading of COH-3/4, but not REC-8, to pachytene axial elements. We have also added a new Figure S1B with a diagram explaining the different regions that were used to quantify signal intensity in the FRAP experiments.

We have also included images of the FRAP experiments in the *wapl-1* mutant background shown in Figure 2A for REC-8::GFP and COH-3::GFP COH-4::GFP (the previous version only showed graphs for these experiments). Images clearly show that removing WAPL-1 reduces COH-3/4 reloading to pachytene axial elements.

Figures 1E and 1F: Signals outside of dotted squares are also decreased after photo-bleaching.

Because of the laser intensity used for the high-resolution images some bleaching of the signal caused during acquisition is unavoidable and this affects all regions of the image, not just the bleached area. All panels are adjusted with the same settings to facilitate comparison of images at different time points.

Figure 2B and 2C: SC-2(GFP) images had a high background in Figure 2B, but not in the bottom right image.Also, images in the control experiments (no photobleaching in Figure 1 and staining in wild type without the tags) and other representative images should be provided in supplement or data storage sites. For image quantification such as intensity measurement, it is critical to show how to normalize the intensity between strains, in some cases with a high background signal, e.g. in Figures 1B 2B, and 2C, COH-4 and REC-8 show two distinct populations for the fluorescence. In Figures 2B, and C, a high background made it hard to quantify the intensity.In addition, the differences between COH-3/4 staining between Figure 1 and Figure 2 are enormous (ca 7x), suggesting that SCC-2::AID is strongly sub-functional. This issue needs to be addressed at the minimum, or better eliminated by using a construct that behaves more like SCC-2.

We have implemented multiple changes in figures and text to clarify the issues mentioned above.

First, we show new examples of raw images of COH-3/4 and REC-8::HA before and after SCC-2::AID::GFP depletion and have also performed experiments at an earlier time point now shown in Figure S2C-E. These confirmed that the effect on the reduction of COH-3/4 staining is weaker at 8 hours post auxin treatment than after 14 hours. As expected, the region of nuclei that failed to load REC-8 during meiotic S-phase is also larger at 14 hours post auxin exposure than at 8 hours. We have included whole germlines examples of these experiments in Figure S2D-E.

Second, REC-8 and COH-3/4 intensity staining in Figure 1B and Figure 2B can’t be directly compared to one another as REC-8 and COH-3/4 were visualised using different antibodies and signal intensity was calculated used different methods. In Figure 1B, all three kleisins were imaged with anti-GFP antibodies to allow direct comparison of REC-8::GFP, COH-3::GFP, and COH-4::GFP intensity at axial elements. On the other hand, experiments shown in Figures 2B-C were performed in strains expressing SCC-2::AID::GFP, and therefore anti-GFP antibodies were used to visualise SCC-2::AID::GFP, while COH-3/4 were visualised with anti-COH-3/4 antibodies and REC-8::HA was visualised using anti-HA antibodies. Because of the use of different primary and secondary antibodies, the fluorescence intensity values are not directly comparable between Figure 1B and Figures 2B-C, or between anti-COH-3/4 staining and anti-HA (REC-8) staining in Figure 2B. The method used for signal-intensity quantification is also different between Figure 1B and Figures 2B-C, with Figure 1B using axis intensity measured in half-nucleus sum projections while Figure 2 uses mean intensity of whole nucleus projections. The reason for using these two different methods is that the goal of the experiment in Figure 1B was to show the relative intensity of REC-8::GFP, COH-3::GFP, and COH-4::GFP at axial elements, which was possible as all three kleisins displayed clear axis staining. In contrast, the weaking of COH-3/4 signal at axial elements following SCC-2 depletion meant that whole nucleus projections offer a more accurate representation of signal intensity. We apologise for not clearly explaining this on the previous version. We now describe the rationale for both signal quantification methods in the text (lines 97-100 and 113-115) and have also added diagrams in Figures 1B (axis sum intensity) and 1D (whole nucleus mean intensity) to describe these two approaches.

Third, regarding the functionality of SCC-2::AID::GFP, we now show that crossover formation is normal in worms homozygous for *scc-2::AID::GFP* and *TIR1* (Figure S2B).

2. For the AID experiments (Figure 2), it is critical to understand how fast SCC-2 is depleted and whether depletion is complete and therefore, for how long the depletion lasts in the various cell types. If possible, please show how much depletion of the protein was achieved by western blotting.

As mentioned above, we now include experiments showing depletion of SCC-2::AID::GFP at an earlier time point (8 hours). Anti-GFP staining confirms efficient depletion of SCC-2::AID::GFP at this point and visualisation of REC-8::HA and COH-3/4 in whole germlines now shown in Figures S2D-E confirms that SCC-2 function is lacking at both time points. We have also included cartoons in Figures 2B-C to explain how nuclei progress through the germline. Because nuclei progress moving at about 1 row per hour and take about 36 hours to transition from early meiosis to late pachytene, using 8 and 14 hours time points means that we can clearly assess the effect of depleting SCC-2::AID::GFP from pachytene nuclei. We appreciate the suggestion to monitor SCC-2::AID::GFP depletion using western blots (WB), however, monitoring protein levels by WB requires using whole worm lysates and as the promoter driving TIR1 expression (required for auxin-meditated protein degradation) is germline specific, SCC-2::GFP::AID is still present in somatic tissues after auxin treatment. Therefore, we can’t use western blotting to monitor SCC-2::AID::GFP depletion. However, as mentioned above, the impairment of REC-8::HA loading in early meiosis nuclei seen after 8 and 14 hours of auxin treatment, together with the loss of GFP signals, clearly confirms efficient depletion of SCC-2::AID::GFP.

Is the loading of COH-3/4 dependent on SCC-2?

We have confirmed that the initial loading of COH-3/4 to chromosomes depends on SCC-2 by performing anti-COH-3/4 staining in germlines of homozygous *scc-2* mutant worms (Figure S2A).

A gonad, like that in Figure 2C for REC-8, could be shown and analyzed for COH-3/4 as well.

We now show full germlines stained with anti-COH-3/4 antibodies after 8 and 14 hours of SCC-2::AID::GFP depletion (Figure S2D). We also show full germlines visualising REC-8::HA after 8 and 14 hours of auxin treatment (Figure S2E).

3. The results showing the role of COH-3/4 in meiotic DSB repair are very weak at present. Additional experiments for DAPI and RAD-51 focus counting in spo-11 coh-3/4 with irradiation could help, introducing a second mutation (e.g. rad-54), which stalls any repair to eliminate the possibility of "rapid" repair.

Our main conclusion from the DSB repair experiments is that COH-3/4 play a minor role in DSB repair compared to REC-8 complexes. We have further confirmed this by comparing RAD-51 staining in late pachytene nuclei of mutants lacking SC, which prevents inter-homologue DSB repair and therefore induces accumulation of RAD-51 foci, and either REC-8 or COH-3/4. While mutants lacking SC and REC-8 accumulate huge numbers RAD-51 foci even in diakinesis oocytes, RAD-51 foci disappear during late pachytene-diplotene of mutants lacking SC and COH-3/4. This result is consistent with our proposal that REC-8 cohesin promotes inter-sister DSB repair that is likely linked to its role in providing SCC.

4. One concern about the manuscript is how much different the contents of the paper are from the authors' previous publication (Crawley et al. eLife 2014; Castellano-Pozo et al. Nature Commun, 2020). It is essential to clarify what is new in this paper and what is confirmation of the previous works (or maybe correction of the work).

The current manuscript elaborates on some of the findings and experimental approaches reported in our previous studies to reveal novel and important aspects of meiotic cohesin function. First, using FRAP, we show that REC-8 and COH-3/4 complexes display differential dynamics in pachytene nuclei and then show that this process is regulated by WAPL-1 and SCC-2. Our analysis of *wapl-1* mutants reported in Crawley et al. 2016 showed that WAPL-1 restricted COH-3/4 loading at early meiotic prophase, but that study did not address the question of whether REC-8 and COH-3/4 associated dynamically with meiotic chromosomes and whether SCC-2 or WAPL-1 affected this process in pachytene nuclei. Our finding that SCC-2 is required for maintaining the levels of COH-3/4 cohesin in pachytene nuclei has important implications to understand meiotic chromosome organisation. As SCC2 is required for the loop extrusion activity of cohesin, we now propose that COH-3/4 complexes may carry out loop extrusion in pachytene nuclei. Second, using TEV-tagged versions of REC-8 and COH-3/4 we unequivocally establish that SCC is uniquely provided by REC-8 complexes in metaphase I oocytes. Although we and others have suggested previously that this may be the case based on observations made in different meiotic mutants, a convincing demonstration of this possibility was lacking and an alternative model proposing that COH-3/4 cohesin do provide SCC had been proposed (Severson et al. 2014). Although the Castellano-Pozo et al. 2020 manuscript described the creation of TEV-tagged versions of REC-8 and COH-3/4, the questions of which complexes provide SCC was not addressed in that study. Our current findings lead us to propose a model in which different populations of cohesin (defined by kleisin identity in worms) perform SCC and loop extrusion. This is an important conceptual advance that is likely a conserved feature of meiosis. Third, by introducing DSBs by IR in pachytene nuclei we demonstrate that low abundance REC-8 complexes play a much more prominent role in DSB repair than highly-abundant COH-3/4 complexes and suggest that this activity is coupled to REC-8’s role in SCC. The role of REC-8 and COH-3/4 complexes in repairing DSBs introduced in pachytene nuclei had not been explored in our previous manuscripts. We have introduced changes in the text and figures (including a model) to highlight the main findings of the current manuscript.

The suggestion that our current work contradicts results from our 2020 Nature Communications manuscript is derived from comments made in the public review of reviewer 3. This reviewer refers to data shown in Figure 1C saying that we used a degron system to degrade COH-3/4 and REC-8 from pachytene nuclei: “The current study uses a degron instead of TEV and SIM to revisit the same result. This time, degradation of COH-3/4 alone, but not of Rec8 alone completely eliminates axial elements. It seems that, if the conclusion is now correct, the previous headline must be incorrect, showing that more care has to be taken in the conclusions.”

This statement is factually incorrect, the SIM images shown in Figure 1C correspond to *rec-8* and *coh-3 coh-4* double mutants (as indicated in main text and figure legend) and therefore to germlines lacking REC-8 or COH-3/4 from the onset of meiosis. In contrast, in the Castellano-Pozo et al. 2020 study REC-8 or COH-3/4 were removed from pachytene chromosomes using the TEV approach following normal chromosome morphogenesis at meiosis onset. This experiment was performed to specifically address how kleisin removal in nuclei at the pachytene stage impacted on meiotic progression, this experiment is not revisited in the current manuscript. In addition to this, Figure 1C does not show that lack of COH-3/4 “completely eliminates axial elements”, as stated by the reviewer, but rather that “SMC-1::GFP signals appeared as discontinuous weak signals in pachytene nuclei” (see description of this result in lines 103-104 of original submission). Moreover, as requested, we now show that axial elements (visualised by HORMAD protein HIM-3) are also discontinuous in *coh-3 coh-4* double mutants but form linear structures in *rec-8* mutants, as also reported by others. Thus, findings in the current manuscript are entirely consistent with those of the Castellano-Pozo et al. 2020 study, where we reported that staining of HORMADs (used to visualise axial elements) became weaker and more discontinuous following removal of COH-3/4 than REC-8 from pachytene axial elements (this observation is also mentioned in lines 96-97 of original submission).

Reviewer #1 (Recommendations for the authors):The paper elegantly reveals the role of REC-8 (but not of COH-3/4) in sister chromatid cohesion (SCC) in diakinesis and metaphase I. Importantly, REC-8 coheisin binds to meiotic chromosomes stably while COH-3/4-cohesin is more dynamic, probably accompanied by dissociation and association, which is controlled by cohesin regulators, SCC-2 and WAPL-1. The experiments in this paper were mostly conducted in a good shaper and most of the conclusions were supported by the data. The paper is clearly written and the conclusion is straightforward. On the other hand, there are some concerns about the data in the paper, which should be addressed prior to publication.

We thank the reviewer for their overall support of our manuscript.

1. The results shown by FRAP experiments (Figures 1 and 2A) were not convincing. In *C. elegans* meiosis, chromosomes show motion driven by microtubule-dynein, which clearly affects the recovery of FRAP signals. It is best to perform FRAP in the absence of the motion in the Dynein mutants such as the dlc-1 mutant. Alternatively, the authors must argue that the motion did not affect their conclusion in a logical manner. Moreover, Figures 1E and 1F show peculiar behavior of signals. In the outside of the dotted square, GFP/mCherry signals diminished and GFP signals were recovered in Figure 1E (0 versus 10 min).

As mentioned in the response to main comments in page 1 (Figure 1D section) we now include new images and diagrams of the FRAP experiments (see Figures 1E-F, 2A, and S1B). Regarding the issue of chromosome active chromosome movements during early prophase: these occur during the leptotene and zygotene stages, but our FRAP images correspond to mid pachytene nuclei, which don’t display this movement. Some level of nuclear movement is unavoidable when imaging pachytene nuclei in the germline of a live worm. In all FRAP experiments the focal plane in Z was adjusted to keep the bleached region in focus at all time points. We have readjusted the panels in Figures 1G-H (1E-F in previous version) to ensure that levels are equally adjusted in all panels. As focal plane was adjusted to ensure that the bleached area was in focus at all time points, differences in intensity outside this area can also reflect slight changes in focus on chromosomes outside the bleached area due to movements.

2. Figure 2A analyzed the effect of the wapl-1 (Wapl) mutation on Coh-3 dynamics and showed that the Wapl-1 promotes the re-association of Coh-3/4 on chromosomes. However, the authors did not discuss this in a reasonable way. Given that Wapl-1 promote the dissociation of the COH-3/4 cohesin from meiotic chromosomes (previously shown by the authors' group), this result suggests either that Wapl-1 has a new function to promote the association of the cohesin to the chromosomes or that WAPL-1 could remove COH-3/4 cohesin for its re-association. This should be discussed or addressed by analyzing wapl-1 scc-2 double mutants.

As requested by the reviewer we have now depleted SCC-2 from pachytene nuclei of worms lacking WAPL-1 (*wapl-1* RNAi) (Figure 2C). This experiment confirms that SCC-2 is required to sustain normal levels of COH-3/4 cohesin on pachytene nuclei. We now clearly state that WAPL-1 and SCC-2 display antagonistic activities in pachytene nuclei, with WAPL-1 acting to promote the creation of a soluble pool of COH-3/4 cohesin that can be loaded onto pachytene chromosomes by SCC-2. This is also reflected in the new model shown on Figure 5.

3. Since the authors had nice antibodies against COH-3/4 for immuno-staining some experiments should be done in double-staining for REC-8::HA and COH-3/4 in as Figure 2B right and Figure 2C-in Figure 2C. Moreover, in addition to Figure 2B let, it is essential to show the localization of COH-3/4 under SCC-2 depletion in "whole" gonads like in Figure 2C.

Performing the co-staining of REC-8::HA and COH-3/4 is technically challenging as those experiments will actually become a quadruple staining including: anti-HA, anti-COH-3/4, anti-GFP (to monitor depletion of SCC-2::AID::GFP), and DAPI. This will force us to image either REC-8 or COH-3/4 with secondary antibodies that emit in the far-red channel, which in our hands provides weaker signal. Because of this limitation we decided to image REC-8::HA and COH-3/4 on separate experiments using secondary antibodies that allow imaging on the red channel.

As requested, we now show full germlines at the different time points after SCC-2::AID::GFP depletion showing REC-8::HA and COH-3/4 staining (Figures S2D-E).

4. If the WAPL-1 controls the localization of COH-3/4, but not of REC-8, GFP-intensity measurement should show more COH-3/4-GFP signals in the wapl-1 mutant than the control, but the same amount of REC-8-GFP by measuring the intensity as shown in Figure 1B. This should be shown.

The fact that COH-3/4 cohesin is the main targets of WAPL-1 has been shown by us (Crawley et al. 2016) and confirmed by others (Hernandez et al. 2018 PLoS Genet 14: e1007382; Yu et al. 2023 Nat Struct Mol Biol 30: 436). In the new experiment shown on Figure 2C we also show increased levels of COH-3/4 in wapl-1RNAi worms. Therefore, we do not think it necessary to include this again in another figure.

5. It is nice for the authors to show a more detailed structure of lobed bivalent in Figure 3A, by analyzing the structure with a super-resolution fluorescent microscope. For readers, it is nice to show a schematic drawing of the structure.

In our opinion, the magnified insets provided in Figure 3A provide clear examples of partial or full separation of sister chromatids after TEV-mediated REC-8 removal from diakinesis oocytes. We do not think that using SR microscopy, which will involve a substantial amount of additional work, will provide further insight. We provide additional examples of the effect of REC-8 removal (TEV or auxin) in diakinesis oocytes in Figures S3A-B, which confirm partial disassembly of diakinesis bivalents following REC-8 removal.

6. Given the contribution of cohesin in the formation of meiosis-specific chromosome axis structure, the authors showed and/or mentioned the role of REC-8 and COH-3/4 in axis structure by looking at the localization of axis proteins such as HORMD proteins.

As requested, we now include staining of HORMAD protein HIM-3 in WT controls, *rec-8* single, and *coh-3 coh-4* double mutants in Figure S1A.

7. Reproducibility: In some experiments such as Figure 2B, multiple nuclei in a single worm were measured. It is critical to show the results for independent worms.

We apologise for not having explained this more clearly, but in all experiments where nuclei were quantified germlines from multiple worms were included. In the case of Figure 1B, nuclei from 3 different germlines (per genotype) were included. This is now indicated in the figure legend.

Reviewer #3 (Recommendations for the authors):Technical:Figure 1Figure 1D is not convincing at all.REC-8::GFP – The object that is bleached is discontinuous with the other chromosomes. The fact that it disappeared is therefore no evidence that it was bleached without subsequent recovery. It is not clear what it was in the first place.COH-3/4::GFP – Here the bleaching of a part of a chromosome was clearly successful, but the pictures do not convince me of recovery after 10 or even 20min. Instead, there is a substantial background that obscures a possible slight recovery.

We have included new images of the FRAP experiments, including a zoom in of the bleached area that clearly illustrates reloading of COH-3/4::GFP, but not REC-8::GFP.

The analysis seems not useful, if it is so difficult, to find even one "representative" case. If the authors manage to repeat the experiment more convincingly, it might be helpful to measure values for the background and its changes over time separately (dispersed signal between the actual chromosomes axes).

Figure S1B now shows a diagram indicating the different regions of interest that were used for the FRAP experiments, showing that overall nuclear signal was used to normalise changes in overall signal over time.

Figure 1 E. Only a part of the nucleus should be bleached to study redistribution of label – however at 0 min after bleaching REC-8::GFP to the left outside of the squared box is dramatically bleached. This suggests, that the full nucleus was bleached instead of just a part. This in turn dramatically disturbs the equilibration kinetics.

The bleached area is indicated by the dashed-line square, but because of the laser intensity used for the high-resolution images some bleaching of the signal caused during acquisition is unavoidable and this affects all regions of the image, not just the bleached area.

The fact that the signal outside the bleaching field is actually back at 10 min is in contrast to the conclusions the authors derive.If the signal came back, this would speak for clear dynamics of the REC-8 protein.In addition, the chromosomes to the right of the bleaching area have a different shape – raising the possibility of movement, or an altered focal plane.

We have readjusted the panels in Figures 1G-H (1E-F in previous version) to ensure that levels are equally adjusted in all panels. As focal plane was adjusted to ensure that the bleached area was in focus at all time points, differences in intensity outside this area can also reflect slight changes in focus on chromosomes outside the bleached area due to movements. Imaging is performed in pachytene nuclei inside the germline of a live worm, some level of movement is unavoidable.

(see also typo on Y-axis label)

This is now corrected.

line 147: " the focal plane was adjusted to follow the indicated region of the bleached axial element, while other regions can be out of focus."I think it may be difficult to focus on the bleached element, which is invisible because it is bleached. If the nuclei move very strongly it may be impossible to do meaningful FRAP without a control that allows to correct for the movement.

As the area being imaged is much larger than the bleached area it is easy to focus on the bleached area by making sure that signal of the axes on the boundaries of the bleached area are in focus.

Figure 1 F. Similar to E the full nuclei seem to have been bleached. Again chromosome parts outside the bleaching area recover.

See comments above regarding Figure 1E.

In summary, the documentation of the bleaching experiments is not convincing. I concede, that it is difficult.Figure 2My main problem with Figure 2 is that COH-3/4 untreated is now weak, punctate, or patchy and nothing like COH-3/4 in Figure 1B.The authors write that COH-3/4 is 3.5 times stronger than REC-8 (l95) – yet COH-3/4 axes appear weaker than REC-8 axes and discontinuous.The quantification shows an intensity average of 600 for COH-3/4 and double (1200) for REC-8. This varies by 2x3.5=7 fold from the statement on (l95).The issue needs to be addressed properly. Should scc-2::AID cause the (7 fold)! reduction of COH-3/4 levels on chromatin, then scc-2::AID is a strongly sub-functional scc-2 version. It would be ideal, to change to a construct with better performance, but at the very minimum, those constitutive scc-2::AID defects should be directly addressed and described.

REC-8 and COH-3/4 intensity staining in Figure 1B and Figure 2B can’t be directly compared to one another as REC-8 and COH-3/4 were visualised using different antibodies and signal intensity was calculated used different methods. In Figure 1B, all three kleisins were imaged with anti-GFP antibodies to allow direct comparison of REC-8::GFP, COH-3::GFP, and COH-4::GFP intensity at axial elements. On the other hand, experiments shown in Figures 2B-C were performed in strains expressing SCC-2::AID::GFP, and therefore anti-GFP antibodies were used to visualise SCC-2::AID::GFP, while COH-3/4 were visualised with anti-COH-3/4 antibodies and REC-8::HA was visualised using anti-HA antibodies. Because of the use of different primary and secondary antibodies, the fluorescence intensity values are not directly comparable between Figure 1B and Figures 2B-C, or between anti-COH-3/4 staining and anti-HA (REC-8) staining in Figure 2B. The method used for signal-intensity quantification is also different between Figure 1B and Figures 2B-C, with Figure 1B using axis intensity measured in half-nucleus sum projections while Figure 2 uses mean intensity of whole nucleus projections. The reason for using these two different methods is that the goal of the experiment in Figure 1B was to show the relative intensity of REC-8::GFP, COH-3::GFP, and COH-4::GFP at axial elements, which was possible as all three kleisins displayed clear axis staining. In contrast, the weaking of COH-3/4 signal at axial elements following SCC-2 depletion meant that whole nucleus projections offer a more accurate representation of signal intensity. We apologise for not clearly explaining this on the previous version. We now describe the rationale for both signal quantification methods in the text (lines 97-100 and 113-115) and have also added diagrams in Figures 1B (axis sum intensity) and 1D (whole nucleus mean intensity) to describe these two approaches.

Also, SCC-2 staining does suffer from high background. Therefore it is close to impossible to judge the amount of SCC-2 after depletion. If cells are treated with auxin for 14 hours, secondary effects may arise.It would be helpful to perform Western blotting, demonstrating how late after the addition of auxin SCC-2 disappears, and to which levels it is reduced. At the same time, it can be used as a control to test whether overall cellular levels of COH-3/4 and REC-8 areunaffected at least by the scc-2::AID construct in the absence of auxin. Unfortunately, the moving window shown in Figure 2C right panel, cannot be captured by such an analysis.

We now show new images of SCC-2::AID::GFP depletion and include experiments after 8 and 14 hours of auxin exposure. Anti-GFP staining confirms efficient depletion of SCC-2::AID::GFP at both points and visualisation of REC-8::HA and COH-3/4 in whole germlines now shown in Figures S2D-E confirms that SCC-2 function is also lacking at both time points. We have also included cartoons in Figures 2B-C to explain how nuclei progress through the germline. Because nuclei progress moving at about 1 row per hour and take about 36 hours to transition from early meiosis to late pachytene, using 8 and 14 hours time points means that we can clearly assess the effect of depleting SCC-2::AID::GFP from pachytene nuclei. We appreciate the suggestion to monitor SCC-2::AID::GFP depletion using western blots (WB), however, monitoring protein levels by WB requires using whole worm lysates and as the promoter driving TIR1 expression (required for auxin-meditated protein degradation) is germline specific, SCC-2::GFP::AID is still present in somatic tissues after auxin treatment. Therefore, we can’t use western blotting to monitor SCC-2::AID::GFP depletion. However, as mentioned above, the impairment of REC-8::HA loading in early meiosis nuclei seen after 8 and 14 hours of auxin treatment, together with the loss of GFP signals, clearly confirms efficient depletion of SCC-2::AID::GFP.

Figure 3312-314 „Since SCC is established during meiotic S-phase and must persist until the meiotic divisions, and given that REC-8 provides SCC during metaphase I, we hypothesised that SCC is provided by REC-8 cohesin throughout the meiotic prophase. We tested this possibility by monitoring SCC at the chromosomal end of the X chromosomes bound by the HIM-8 protein"The HIM-3 experiment tests, whether REC-8 is required to maintain SCC on the X chromosome at some point in prophase if TEV is induced at some point in prophase.It does not show the requirement „throughout" prophase.

We have modified to text to say that separation of sister chromatids of the X chromosome following TEV-mediated removal REC-8 is consistent with SCC being provided by REC-8 complexes in pachytene.

Figure 4398-400 „Chromosome fragments were largely lacking from diakinesis oocytes of wild-type worms exposed to 100 Gy of IR, evidencing efficient DSB repair (Figure 4A)."Could the author indicate, at which time after irradiation the observation was made in diakinesis? How long did the cells have time to repair? What was the stage of the cells during irradiation?

Diakinesis nuclei were analysed 26 hours after irradiation, this is now indicated on the main text as well as on figure legend. This time point allows us to analyse the effect of DSBs that were introduced on nuclei that were already at the pachytene stage at the time of irradiation.

The experiment shown in Figure 4 does not make a very strong additional point for a direct role of REC-8 in DSB repair.Fragmentation after irradiation is an expected consequence after the loss of SCC, thus if REC-8 depletion results in loss of SCC this observation per se could be explained by the above-described defect of REC-8.4A is complemented by the experiment in Figure 4B, which shows that the kinetics of disappearance of RAD-51 foci are at least slower in rec-8 mutants. Again though, this could be a direct consequence of the lack of SCC delaying homology search, rather than proving a more direct role of REC-8 in DSB repair.

We agree with the reviewer that our experiments suggest that the main reason why REC-8, despite its lower abundance, has a more prominent in DSB repair than COH-3/4 is its ability to provide SCC. Our experiments were performed to compare the DSB repair capacity of both types of meiotic cohesin, not to determine whether the role of REC-8 in DSB repair is different from its role in SCC. We clearly state at the end of this section that “Our results suggest that similar to yeast Scc1 cohesin during mitotic G2, the role of REC-8 in DSB repair during meiotic prophase is mechanistically coupled to its ability to provide SCC.

Conclusions435-438 The abundance shown in Figure 1 is not reflected in Figure 2. This creates some uncertainty about this relationship.Is the scc-2::AID sub-functional?

Please see our response above to comments about Figure 2, which explains in detail why the intensity values in graphs shown in Figures 1B and 2A can not be directly compared (use of the different primary and secondary antibodies as well different quantification methods).